# Efficient Fine-Tuning via Behavior-Guided Spectral Alignment

## Abstract

Parameter-Efficient Fine-Tuning (PEFT) has become a practical approach for adapting large vision models with limited data and computational resources. However, existing PEFT methods primarily focus on where to inject trainable parameters, providing little guidance on how internal representations evolve during adaptation. This often results in *a passive fine-tuning process* that lacks explicit alignment with the target task's structure, especially in settings with limited data or diverse tasks. We propose Behavior-Aligned Fine-Tuning (BAFT), a simple, parameter-free and teacher-free method that introduces behavioral constraints during fine-tuning without changing the model architecture. BAFT extracts the relational structure of model predictions, capturing how samples relate in the output space, and aligns it with intermediate feature representations by minimizing the distance between their cosine similarity matrices. This alignment acts as a lightweight, task-aware regularizer that guides internal representations to better reflect the decision structure of the target task. BAFT requires no additional trainable parameters, adds minimal overhead, and integrates seamlessly with a wide range of PEFT methods including LoRA, AdaptFormer, Bi-LoRA, and Bi-AdaptFormer. On VTAB-1k and few-shot fine-grained classification benchmarks, BAFT consistently improves performance compared to strong PEFT baselines. Analyses of gradient behavior, spectral alignment, and attention dynamics further demonstrate how BAFT promotes more structured and task-aligned representations. By transforming output-space behavior into *actionable training signals*, BAFT reframes fine-tuning as an active and guided process. This work offers a novel and principled direction for advancing parameter-efficient model adaptation.

## 1 Introduction

Large-scale vision models, particularly Vision Transformers (ViTs), have become a cornerstone of modern computer vision, achieving strong performance across a wide range of tasks (Dosovitskiy et al., 2020; Kirillov et al., 2023; Liu et al., 2021; Zhu et al., 2024; Ding & Wang, 2024; 2025). However, adapting these pretrained models to new downstream tasks typically requires significant computational resources and fine-tuning effort. To mitigate these costs, Parameter-Efficient Fine-Tuning (PEFT) methods have emerged as a practical alternative (Houlsby et al., 2019). These techniques adapt large models by inserting additional learnable parameters (Jia et al., 2022), lightweight task-specific modules, such as adapters (Chen et al., 2022; Jie et al., 2024; Jie & Deng, 2023; Karimi Mahabadi et al., 2021; Jie et al., 2023) or attention decomposition (LoRA) (Hu et al., 2022), while keeping most of the backbone frozen. By dramatically reducing the number of trainable parameters, PEFT methods have become widely adopted in low-resource and rapid adaptation scenarios.

Despite their success, current PEFT methods focus mainly on the mechanics of parameter injection, where and how to insert new modules, while paying little attention to the model's behavioral dynamics during adaptation. In most cases, fine-tuning is treated as *a passive process*: learnable components are trained to improve task performance, but the evolving internal representations are not explicitly guided to reflect the target task's decision patterns. This lack of behavioral oversight means that adaptation may proceed in arbitrary or inefficient directions, especially in low-data or task-diverse conditions (Wang et al., 2024b). As a result, fine-tuning remains a largely unstructured and opaque process, with no mechanism to ensure that the model's internal learning trajectory aligns with what the task actually requires.

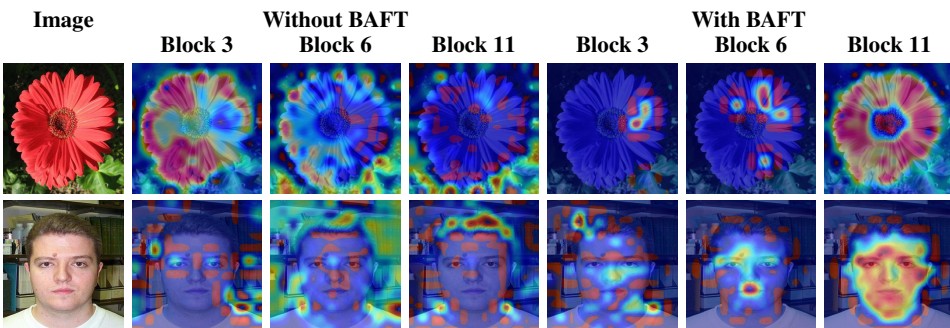

Figure 1: Grad-CAM (Selvaraju et al., 2016) visualizations of the 3rd, 6th and final block of Bi-AdaptFormer fine-tuned with and without BAFT on two VTAB-1k datasets. Top row: OxfordFlowers; bottom row: Caltech101. BAFT leads to sharper, more task-relevant attention focus.

In this work, we introduce Behavior-Aligned Fine-Tuning (BAFT), a simple method that brings structure and task awareness into the fine-tuning process. Our key insight is that the prediction scores generated during training already contain rich relational information about how the model perceives sample similarities and decision boundaries. For example, if two inputs yield similar softmax distributions, they are likely close in the model's output space. We use this observation by dynamically extracting batch-level prediction structures and encouraging the model's intermediate feature representations (*e.g.*, class token embeddings in ViTs) to reflect those same relationships. This forms a lightweight, training-time constraint that gently steers the model's internal adaptation to mirror its evolving task-specific behavior.

BAFT introduces no additional parameters, and can be plugged into any PEFT method. Rather than modifying the model's architecture, BAFT shapes its learning behavior by turning existing training signals into a form of behavioral guidance, effectively transforming fine-tuning from a parameter-centric procedure into a behavior-aware process. We evaluate BAFT on the VTAB-1k (Zhai et al., 2019) benchmark, which spans 19 diverse vision tasks, as well as in few-shot learning settings that challenge the model's ability to adapt under limited supervision. We apply BAFT to a range of strong PEFT baselines, including LoRA (Hu et al., 2022), AdaptFormer (Chen et al., 2022), Bi-LoRA (Jie et al., 2023), and Bi-AdaptFormer (Jie et al., 2023), and consistently observe performance improvements across the board. As shown in Fig. 1, models trained with BAFT consistently produce more focused and semantically meaningful activation maps. For example, in the OxfordFlowers image, BAFT sharpens attention around the flower's petals, whereas the baseline shows diffuse or misaligned focus. In the Caltech101 image, BAFT shows strong, focused attention on key areas (eyes, nose and mouth). These results show BAFT's effectiveness across diverse visual tasks.

To our knowledge, this is the first work to introduce a parameter-free, behavior-guided mechanism that aligns intermediate representations with task-specific prediction patterns during fine-tuning. By turning the model's own output structure into training-time behavioral guidance, BAFT makes fine-tuning not only efficient, but also targeted, adaptive, and aligned with the model's decision-making behavior. Our **contributions** are summarized as follows:

i. We *identify a key limitation* in existing PEFT methods: the lack of task-aware guidance over how internal representations evolve during fine-tuning.

ii. We propose Behavior-Aligned Fine-Tuning (BAFT), *a parameter-free, plug-and-play method* that uses prediction-space relationships to guide internal feature adaptation dynamically.

iii. We show *the effectiveness of BAFT* across VTAB-1k and multiple few-shot scenarios, consistently improving performance over state-of-the-art PEFT techniques with no additional parameters or architectural changes.

## 2 RELATED WORKS

**Parameter-efficient fine-tuning (PEFT).** PEFT methods aim to adapt large pretrained models to downstream tasks by updating only a small subset of parameters, thereby drastically reducing the

memory footprint and computational cost compared to full fine-tuning. To reduce the number of fine-tuned parameters, BitFit (Zaken et al., 2021) fine-tunes the bias terms and freezes most of the network. Visual Prompt Tuning(VPT) and its variants (Han et al., 2023; Zeng et al., 2024; Chen et al., 2025; Raj et al., 2025) also introduces learnable prompts to each layer of ViT (Jia et al., 2022). However, because the computational cost of self-attention scales quadratically with input length, prompt-based methods tend to be less efficient than the original network in terms of computation. As such prominent approaches such as adapter-based methods (Chen et al., 2022; Luo et al., 2023; Karimi Mahabadi et al., 2021; He et al., 2023) including AdaptFormer (Chen et al., 2022) inject lightweight trainable modules at various depths of the network, and low-rank update methods such as LoRA (Hu et al., 2022), which optimize low-rank decompositions of weight matrices. Other methods have also explored automatically combining multiple methods (Chavan et al., 2023; Zhang et al., 2024). Recently, other methods have also explored several techniques towards extreme parameter efficiency and memory efficiency (Jie & Deng, 2023; Jie et al., 2023; Fu et al., 2024; Zhang et al., 2020). For instance, FACT (Jie & Deng, 2023) employs a tensor-decomposition framework to store changes in the model's weights, whiles Bi-AdaptFormer (Jie et al., 2023) proposes a low-bit adapter to reduce precision redundancy.While these methods differ in architectural design and parameter injection strategies, they primarily view adaptation as a structural optimization problem, focusing on where and how to insert tunable modules. Crucially, they offer limited insight or control over the behavioral dynamics of model adaptation, that is, how internal representations should evolve during fine-tuning to reflect task-specific decision boundaries. Our method addresses this critical gap by introducing a behavioral-level constraint that dynamically aligns intermediate representations with the evolving task structure, without requiring architectural changes or additional parameters. This makes it a natural complement to all existing PEFT frameworks, enhancing their effectiveness through principled behavioral guidance.

**Regularization and behavioral constraints in fine-tuning.** Regularization techniques have long been used to stabilize fine-tuning, particularly under low-data regimes. Approaches like L2-SP (Xuhong et al., 2018) and Elastic Weight Consolidation (Kirkpatrick et al., 2017) penalize large deviations from pretrained weights to prevent catastrophic forgetting. Other works encourage similarity between source and target domain features (Liu et al., 2020; Jiang et al., 2022) by imposing losses on intermediate layers or feature distributions, promoting better transferability. However, most existing regularization methods are static and global: they apply uniform constraints throughout training and lack adaptation to the specific structure of the target task or the dynamics within each training batch. Moreover, these techniques were generally developed for full fine-tuning scenarios and may not translate well to PEFT, where only a small fraction of parameters are updated. In contrast, our method introduces a lightweight, dynamic, and batch-level regularization strategy that exploits the model's own prediction scores to capture a soft relational structure among samples within each mini-batch. By explicitly aligning internal features with this evolving prediction-based structure, we impose an adaptive, task-aware behavioral constraint that shapes the model's learning trajectory in real time. This approach complements traditional weight-based regularizers by focusing on guiding the representational dynamics through prediction-driven feedback, rather than constraining parameter shifts.

**Output-feature alignment and structural supervision.** Prior work in related areas has explored aligning output distributions with internal representations, often in contexts like knowledge distillation (Hinton et al., 2015) or self-supervised learning (Chen et al., 2020; Wang et al., 2024a). Such methods typically rely on a fixed teacher model providing soft targets, or use contrastive objectives to align features across multiple views or modalities. Our approach differs fundamentally in that it requires no external teacher or multi-view data. Instead, it uses on-the-fly relational structures derived purely from the model's own batch-level prediction scores. We then guide the alignment of internal features, such as class token embeddings in ViTs, with this dynamically computed structure. This batch-level relational alignment serves as a novel, self-supervised form of structural regularization that naturally integrates with PEFT, since it does not require additional supervision, architectural modifications, or learnable parameters. To our knowledge, this is the first method to use batch-level prediction relationships to guide representation alignment in PEFT. By introducing this new perspective on behavioral guidance, our work opens a promising direction for making fine-tuning not only parameter-efficient but also behaviorally and task-wise aligned.

## 3 METHOD

We propose *a parameter-free behavioral alignment mechanism* that dynamically guides fine-tuning by aligning the model's internal feature geometry with semantic structure expressed in its predictions. Crucially, our method introduces no architectural modifications or additional supervision: it operates entirely by reusing the model's own prediction structure as a *dynamic training signal*. This promotes a representation space that is task-aware, behaviorally consistent, and semantically organized, while preserving the parameter-efficiency of the underlying fine-tuning algorithm. Fig. 2 shows the integration of Behavior-Aligned Fine-Tuning (BAFT) into a PEFT-based transformer.

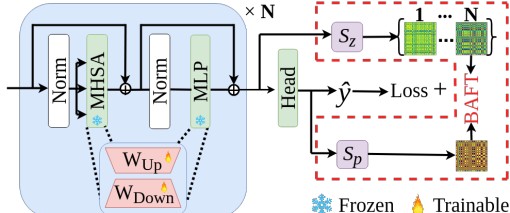

Figure 2: Overview of Behavior-Aligned Fine-Tuning (BAFT) (dashed red) integrated into a PEFT-based transformer. The model uses a standard Transformer with frozen MHSA and MLP layers and trainable low-rank adapters. BAFT computes similarity matrices from predictions ($\mathbf{S}_p$) and features ($\mathbf{S}_z$) at selected layers, then aligns them using a lightweight behavioral loss. This steers feature representations to reflect the model's evolving prediction structure, without adding any learnable parameters.

**Problem setup.** Let $f_{\boldsymbol{\theta}} : \mathcal{X} \to \mathbb{R}^C$ be a neural network with parameters $\boldsymbol{\theta}$, adapted to a target classification task via a PEFT strategy. For an input image $\boldsymbol{x}_i \in \mathcal{X}$, the model yields softmax-normalized predictions $\boldsymbol{p}_i = \mathrm{softmax}(f_{\boldsymbol{\theta}}(\boldsymbol{x}_i)) \in \mathbb{R}^C$ and intermediate feature representations $\boldsymbol{z}_i \in \mathbb{R}^d$ (*e.g.*, from the class token in ViT). We denote a training mini-batch as $\{(\boldsymbol{x}_i, y_i)\}_{i=1}^{B}$ with predictions $\boldsymbol{P} \in \mathbb{R}^{B \times C}$ and features $\boldsymbol{Z} \in \mathbb{R}^{B \times d}$. The central hypothesis of this work is that the relational structure encoded in the prediction space, how confidently and distinctly the model separates different instances, can be repurposed to guide the geometry of the internal representations during fine-tuning, even under PEFT constraints. This alignment serves as a weak form of supervision that constrains the evolution of the feature space in accordance with task semantics.

### 3.1 BEHAVIOR-ALIGNED GUIDANCE

To promote behaviorally aligned, task-aware adaptation during fine-tuning, we introduce a lightweight mechanism that aligns the relational structure of model predictions with intermediate representations. The softmax outputs produced during training inherently encode rich behavioral cues, reflecting how the model perceives similarities between samples in the output space. BAFT uses this emergent signal to guide internal feature evolution, encouraging consistency between prediction behavior and learned representations.

**Behavioral alignment loss.** Given a training batch of size $B$, we define two similarity matrices: one in the prediction space and one in the feature space. Let $\boldsymbol{p}_i$ denote the softmax output of the $i$-th sample, and $\boldsymbol{z}_i$ its corresponding intermediate feature (*e.g.*, the class token in ViT). We compute pairwise cosine similarities across all samples to obtain $\mathbf{S}_p \in \mathbb{R}^{B \times B}$ and $\mathbf{S}_z \in \mathbb{R}^{B \times B}$:

$$\mathbf{S}_p(i,j) = \cos(\boldsymbol{p}_i, \boldsymbol{p}_j), \quad \mathbf{S}_z(i,j) = \cos(\boldsymbol{z}_i, \boldsymbol{z}_j). \quad (1)$$

Cosine similarity is chosen because it reflects angular proximity while being invariant to vector magnitudes, which is particularly important for comparing softmax distributions (bounded and probabilistic) and $\ell_2$-normalized feature embeddings. This choice ensures that alignment is based on semantic directionality rather than scale, which can vary across layers or during optimization without necessarily indicating a meaningful change in relational structure.

Unless stated otherwise, BAFT terms are computed across all model blocks. For each block, intermediate features are extracted to construct a feature similarity matrix. The per-layer BAFT losses are then averaged. By minimizing the distance between feature and output similarity matrices, BAFT encourages the feature space to reflect the evolving structure of the output space:

$$\mathcal{L}_{\mathrm{BAFT}} = \frac{1}{L} \sum_{l=1}^{L} \|\mathbf{S}_z^{(l)} - \mathbf{S}_p\|_F^2, \quad (2)$$

where $\|\cdot\|_F$ is the Frobenius norm and $L$ is the number of layers. This alignment serves as a *behavioral signal*, guiding the representation geometry to reflect the model's evolving task-specific

similarity. The complete training objective becomes:

$$\mathcal{L} = \mathcal{L}_{\mathrm{Sup}} + \lambda \mathcal{L}_{\mathrm{BAFT}},\tag{3}$$

where $\mathcal{L}_{\mathrm{Sup}}$ is the standard supervised loss (*e.g.*, cross-entropy), and $\lambda$ is a tunable scalar that governs the strength of alignment. Importantly, BAFT introduces no additional learnable parameters, no architectural modifications. It can be seamlessly integrated into any existing PEFT method. By dynamically extracting the prediction-space structure and aligning it with internal representations, BAFT provides a parameter-free, task-aware behavioral signal that gently guides model adaptation. Rather than treating fine-tuning as a parameter-centric procedure, BAFT reinterprets it as a behavior-aware process, enhancing both interpretability and effectiveness without sacrificing efficiency.

## 3.2 Behavioral Dynamics via Gradient Analysis

We now formally analyze how the behavioral alignment loss $\mathcal{L}_{\mathrm{BAFT}}$ influences representation learning during fine-tuning. Specifically, we derive its gradient with respect to the intermediate feature vectors and show how it induces structured updates that reflect the task-relevant relational geometry embedded in the model's own prediction space.

For clarity, we assume all vectors are $\ell_2$-normalized (*i.e.*, $\|\boldsymbol{z}_i\| = \|\boldsymbol{p}_i\| = 1$), such that cosine similarity reduces to a dot product. Let $\boldsymbol{Z} \in \mathbb{R}^{B \times d}$ and $\boldsymbol{P} \in \mathbb{R}^{B \times C}$ denote the intermediate feature and prediction matrices, respectively, with each row representing a feature vector $\boldsymbol{z}_i$ or prediction vector $\boldsymbol{p}_i$. We define the discrepancy $\delta_{ij} := \boldsymbol{z}_i^\top \boldsymbol{z}_j - \boldsymbol{p}_i^\top \boldsymbol{p}_j$, allowing us to express the loss as:

$$\mathcal{L}_{\mathrm{BAFT}} = \|\boldsymbol{Z}\boldsymbol{Z}^\top - \boldsymbol{P}\boldsymbol{P}^\top\|_F^2 = \sum_{i,j}\left(\boldsymbol{z}_i^\top \boldsymbol{z}_j - \boldsymbol{p}_i^\top \boldsymbol{p}_j\right)^2 = \sum_{i,j}\delta_{ij}^2.\tag{4}$$

We compute the gradient of $\mathcal{L}_{\mathrm{BAFT}}$ with respect to an arbitrary feature vector $\boldsymbol{z}_k$:

$$\frac{\partial \mathcal{L}_{\mathrm{BAFT}}}{\partial \boldsymbol{z}_k} = \sum_{i,j}\frac{\partial \delta_{ij}^2}{\partial \boldsymbol{z}_k} = 2\sum_{i,j}\delta_{ij} \cdot \frac{\partial(\boldsymbol{z}_i^\top \boldsymbol{z}_j)}{\partial \boldsymbol{z}_k}.\tag{5}$$

Noting that the derivative of $\boldsymbol{z}_i^\top \boldsymbol{z}_j$ w.r.t. $\boldsymbol{z}_k$ is non-zero only when $k = i$ or $k = j$, we obtain:

$$\frac{\partial \mathcal{L}_{\mathrm{BAFT}}}{\partial \boldsymbol{z}_k} = 2\sum_{j=1}^{B}\left(\delta_{kj}\boldsymbol{z}_j + \delta_{jk}\boldsymbol{z}_j\right) = 4\sum_{j=1}^{B}\delta_{kj}\boldsymbol{z}_j = 4\sum_{j=1}^{B}\left(\boldsymbol{z}_k^\top \boldsymbol{z}_j - \boldsymbol{p}_k^\top \boldsymbol{p}_j\right)\boldsymbol{z}_j.\tag{6}$$

Here, we use the fact that $\delta_{kj} = \delta_{jk}$ due to symmetry.

> This gradient shows how each feature vector $\boldsymbol{z}_k$ is iteratively updated based on its relationship to all other vectors in the batch. The term $\boldsymbol{z}_k^\top \boldsymbol{z}_j - \boldsymbol{p}_k^\top \boldsymbol{p}_j$ quantifies the discrepancy between the actual similarity in feature space and the expected similarity as implied by the prediction space. When the feature similarity underestimates the behavioral similarity, *i.e.*, $\boldsymbol{z}_k^\top \boldsymbol{z}_j < \boldsymbol{p}_k^\top \boldsymbol{p}_j$, the vector $\boldsymbol{z}_k$ is pulled toward $\boldsymbol{z}_j$ to better align the feature space with the behavioral signal. Conversely, when feature similarity overestimates the behavioral similarity, $\boldsymbol{z}_k$ is pushed away from $\boldsymbol{z}_j$. Over time, this dynamic promotes the emergence of a feature space in which examples with similar predicted behavior are encoded with more similar representations. The alignment objective reduces disagreement between feature-space and output-space relational structures.

**Relation to contrastive learning.** While this mechanism bears resemblance to contrastive learning, it departs in a key way: it does not require explicit positives or negatives. Instead, the model constructs a soft, evolving similarity graph over the batch using its own predictions. The alignment loss smooths the internal representations over this graph, imposing a relational inductive bias that adapts at each training step. This form of dynamic, task-aware supervision is more flexible and less brittle than conventional contrastive objectives. It also naturally complements PEFT methods, where architectural constraints limit the space of parameter updates. In such settings, our behavioral alignment loss acts as a high-level guide, shaping representation geometry even when the underlying parameter space is restricted.

Table 1: Comparison of state-of-the-art PEFT methods on VTAB-1k, grouped by Natural, Specialized, and Structured datasets. "Average" reports mean accuracy across all groups. * indicates reproduced baselines using original configurations. **Bold** marks our best variant; underline shows the top existing method per dataset. BAFT consistently improves performance without introducing additional parameters, demonstrating its effectiveness and generality.

| | #Param (M) | Natural | | | | | | | Specialized | | | | Structured | | | | | | | | Average |
|---|---|---|---|---|---|---|---|---|---|---|---|---|---|---|---|---|---|---|---|---|---|
| | | Cifar100 | Caltech101 | DTD | Flowers102 | Pets | SVHN | Sun397 | Camelyon | EuroSAT | Resisc45 | Retinopathy | Clevr-Count | Clevr-Dist | DMLab | KITTI-Dist | dSpr-Loc | dSpr-Ori | sNORB-Azim | sNORB-Ele | |
| *Traditional Fine-Tuning* | | | | | | | | | | | | | | | | | | | | | |
| Full | 85.8 | 68.9 | 87.7 | 64.3 | 97.2 | 86.9 | 87.4 | 38.8 | 79.7 | 95.7 | 84.2 | 73.9 | 56.3 | 58.6 | 41.7 | 65.5 | 57.5 | 46.7 | 25.7 | 29.1 | 68.9 |
| Linear | 0 | 64.4 | 85.0 | 63.2 | 97.0 | 86.3 | 36.6 | 51.0 | 78.5 | 87.5 | 68.5 | 74.0 | 34.3 | 30.6 | 33.2 | 55.4 | 12.5 | 20.0 | 9.6 | 19.2 | 57.6 |
| *PETL Methods* | | | | | | | | | | | | | | | | | | | | | |
| BitFit (Zaken et al., 2021) | 0.10 | 72.8 | 87.0 | 59.2 | 97.5 | 85.3 | 59.9 | 51.4 | 78.7 | 91.6 | 72.9 | 69.8 | 61.5 | 55.6 | 32.4 | 55.9 | 66.6 | 40.0 | 15.7 | 25.1 | 65.2 |
| VPT-Deep (Jia et al., 2022) | 0.53 | 78.8 | 90.8 | 65.8 | 98.0 | 88.3 | 78.1 | 49.6 | 81.8 | 96.1 | 83.4 | 68.4 | 68.5 | 60.0 | 46.5 | 72.8 | 73.6 | 47.9 | 32.9 | 37.8 | 72.0 |
| LoRA (Hu et al., 2022) | 0.29 | 67.1 | 91.4 | 69.4 | 98.8 | 90.4 | 85.3 | 54.0 | 84.9 | 95.3 | 84.4 | 73.6 | 82.9 | 69.2 | 49.8 | 78.5 | 75.7 | 47.1 | 31.0 | 44.0 | 74.5 |
| AdaptFormer (Chen et al., 2022) | 0.16 | 70.8 | 91.2 | 70.5 | 99.1 | 90.9 | 86.6 | 54.8 | 83.0 | 95.8 | 84.4 | 76.3 | 81.9 | 64.3 | 49.3 | 80.3 | 76.3 | 45.7 | 31.7 | 41.1 | 74.7 |
| E²VPT (Han et al., 2023) | 0.25 | 78.6 | 89.4 | 67.8 | 98.2 | 88.5 | 85.3 | 52.3 | 82.5 | 96.8 | 84.8 | 73.6 | 71.7 | 61.2 | 47.9 | 75.8 | 80.8 | 48.1 | 31.7 | 41.9 | 73.9 |
| RepAdapter (Luo et al., 2023) | 0.22 | 69.0 | 92.6 | 75.1 | 99.4 | 91.8 | 90.2 | 52.9 | 87.4 | 95.9 | 87.4 | 75.5 | 75.9 | 62.3 | 53.3 | 80.6 | 77.3 | 54.9 | 29.5 | 37.9 | 76.1 |
| Bi-LoRA (Jie et al., 2023) | 1.18 | 72.1 | 91.7 | 71.2 | 99.1 | 91.4 | 90.2 | 55.8 | 87.0 | 95.4 | 85.5 | 75.5 | 83.1 | 64.1 | 52.2 | 81.2 | 86.2 | 53.5 | 36.7 | 44.4 | 76.7 |
| Bi-AdaptFormer (Jie et al., 2023) | 0.59 | 74.1 | 92.4 | 72.1 | 99.3 | 91.6 | 89.0 | 56.3 | 88.2 | 95.2 | 86.0 | 76.2 | 83.9 | 63.9 | 53.0 | 81.4 | 86.2 | 54.8 | 35.2 | 41.3 | 77.0 |
| NOAH (Zhang et al., 2024) | 0.36 | 69.6 | 92.7 | 70.2 | 99.1 | 90.4 | 86.1 | 53.7 | 84.4 | 95.4 | 83.9 | 75.8 | 82.8 | 68.9 | 49.9 | 81.7 | 81.8 | 48.3 | 32.8 | 44.2 | 75.5 |
| RLRR (Dong et al., 2024b) | 0.33 | 75.6 | 92.4 | 72.9 | 99.3 | 91.5 | 89.8 | 57.0 | 86.8 | 95.2 | 85.3 | 75.9 | 79.7 | 64.2 | 53.9 | 82.1 | 83.9 | 53.7 | 33.4 | 43.6 | 76.7 |
| HTA (Dong et al., 2024a) | 0.22 | 79.0 | 92.8 | 77.6 | 99.6 | 92.4 | 89.4 | 55.1 | 88.2 | 96.1 | 89.7 | 76.4 | 84.2 | 61.7 | 53.6 | 82.0 | 85.1 | 53.7 | 33.9 | 47.9 | 75.7 |
| DMLoRA (Fang et al., 2024) | 0.29 | 74.0 | 90.7 | 73.9 | 99.3 | 92.2 | 91.1 | 56.4 | 85.6 | 96.5 | 87.0 | 76.1 | 83.5 | 69.9 | 52.0 | 81.6 | 80.2 | 50.2 | 36.1 | 43.1 | 77.0 |
| AdaptFormer* (Chen et al., 2022) | 0.16 | 74.2 | 93.0 | 73.1 | 99.3 | 91.7 | 88.8 | 56.4 | 88.3 | 95.6 | 84.8 | 75.1 | 84.2 | 64.1 | 53.0 | 81.7 | 85.5 | 55.4 | 35.1 | 40.1 | 76.9 |
| **Ours: AdaptFormer* + BAFT** | 0.16 | 74.3 | 93.0 | 73.1 | 99.4 | 91.9 | 89.0 | 56.5 | 88.5 | 95.7 | 85.0 | 75.7 | 84.6 | 64.3 | 53.1 | 82.3 | 85.8 | 55.5 | 35.3 | 44.1 | 77.3 |
| LoRA* (Hu et al., 2022) | 0.29 | 72.8 | 92.3 | 72.2 | 99.2 | 91.3 | 89.7 | 56.0 | 86.3 | 95.2 | 83.8 | 75.0 | 83.2 | 64.7 | 52.3 | 79.9 | 85.6 | 53.1 | 36.7 | 43.0 | 76.4 |
| **Ours: LoRA* + BAFT** | 0.29 | 73.2 | 92.5 | 72.3 | 99.3 | 91.4 | 90.0 | 56.3 | 86.7 | 95.4 | 84.0 | 75.2 | 83.3 | 64.9 | 52.8 | 82.1 | 85.7 | 53.7 | 37.2 | 43.9 | 76.8 |
| Bi-LoRA* (Jie et al., 2023) | 1.18 | 73.5 | 92.4 | 71.5 | 99.3 | 91.1 | 89.9 | 56.0 | 86.9 | 95.2 | 84.8 | 74.3 | 83.6 | 64.1 | 53.1 | 80.2 | 85.9 | 53.9 | 39.0 | 43.2 | 76.7 |
| **Ours: Bi-LoRA* + BAFT** | 1.18 | 73.6 | 92.3 | 71.7 | 99.5 | 91.4 | 90.1 | 56.4 | 87.6 | 95.7 | 85.1 | 75.2 | 83.7 | 64.5 | 53.2 | 80.5 | 86.3 | 54.4 | 39.2 | 43.4 | 77.1 |
| Bi-AdaptFormer* (Jie et al., 2023) | 0.59 | 74.6 | 92.7 | 72.6 | 99.4 | 91.5 | 90.1 | 56.4 | 87.3 | 95.6 | 85.1 | 74.4 | 84.9 | 64.1 | 53.6 | 82.1 | 87.2 | 55.2 | 35.8 | 40.5 | 77.0 |
| **Ours: Bi-AdaptFormer* + BAFT** | 0.59 | **75.0** | 92.8 | 72.7 | **99.5** | 91.6 | **90.6** | **56.8** | **88.7** | 95.7 | **85.4** | **76.1** | **85.1** | 64.3 | **53.8** | **82.3** | **87.6** | **55.6** | 36.2 | 40.7 | **77.5** |

## 3.3 THEORETICAL INSIGHT: BEHAVIORAL SPECTRAL ALIGNMENT

To understand how the behavioral alignment loss $\mathcal{L}_{\mathrm{BAFT}}$ shapes the representation geometry, we analyze it through the lens of spectral theory. Specifically, we show that minimizing $\mathcal{L}_{\mathrm{BAFT}}$ encourages the internal feature similarity kernel to approximate the task-induced relational structure in the prediction space, thereby aligning their eigenspaces.

**Theorem 1.** *Let $\boldsymbol{Z} \in \mathbb{R}^{B \times d}$ and $\boldsymbol{P} \in \mathbb{R}^{B \times C}$ be matrices whose rows are $\ell_2$-normalized features and prediction logits, respectively. Then the behavioral alignment loss defined in Eq. 4 is minimized if and only if the Gram matrices $\boldsymbol{Z}\boldsymbol{Z}^\top$ and $\boldsymbol{P}\boldsymbol{P}^\top$ are identical. In particular, $\mathcal{L}_{\mathrm{BAFT}}$ promotes alignment of the top eigenvectors of $\boldsymbol{Z}\boldsymbol{Z}^\top$ and $\boldsymbol{P}\boldsymbol{P}^\top$.*

See Appendix A.4 for the proof.

> This result formalizes $\mathcal{L}_{\mathrm{BAFT}}$ as a spectral kernel alignment objective. It encourages the internal similarity matrix $\boldsymbol{K}_Z$ to mirror the relational structure encoded in the model's predictions $\boldsymbol{K}_P$. The alignment is not merely pointwise but structural: the dominant eigenvectors of $\boldsymbol{K}_Z$, which define principal directions in feature space, are shaped to match those of $\boldsymbol{K}_P$, encoding semantic relations in the output space. Importantly, this process operates at the batch level and is inherently relational. Rather than enforcing hard labels or margin-based constraints, the alignment loss encourages smooth, soft pairwise consistency among features, reflecting the evolving behavior of the model. In doing so, it transfers inductive bias from the output space back into the feature space, refining internal geometry in a way that is both data- and task-dependent.

**Relation to multi-view learning.** This spectral alignment is analogous to co-regularization in multi-view learning, where similarity kernels from different modalities are aligned. Here, prediction vectors $\mathbf{p}_i$ and feature vectors $\mathbf{z}_i$ act as two views from the same model. The alignment loss ensures they co-evolve coherently, reinforcing semantic consistency without extra supervision. This provides an effective mechanism for guiding representation learning in constrained settings like PEFT, where capacity is limited and efficient structure transfer is crucial.

## 4 EXPERIMENT

We validate our method by applying it to four leading PEFT techniques: AdaptFormer (Chen et al., 2022), LoRA (Hu et al., 2022), Bi-AdaptFormer (Jie et al., 2023) and Bi-LoRA (Jie et al., 2023), demonstrating broad compatibility and consistent gains.

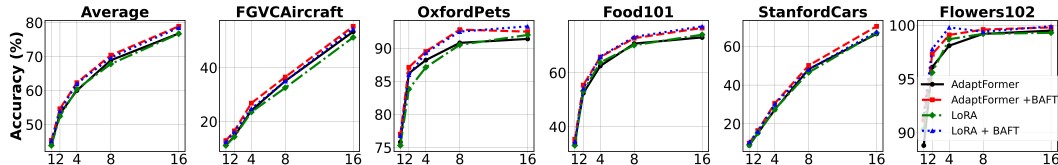

Figure 3: Few-shot learning results on five FGVC datasets across varying shot settings (1, 2, 4, 8, 16). Models with BAFT consistently outperform their baselines, highlighting its effectiveness in low-data regimes. Horizontal axis: number of shots; vertical axis: classification accuracy.

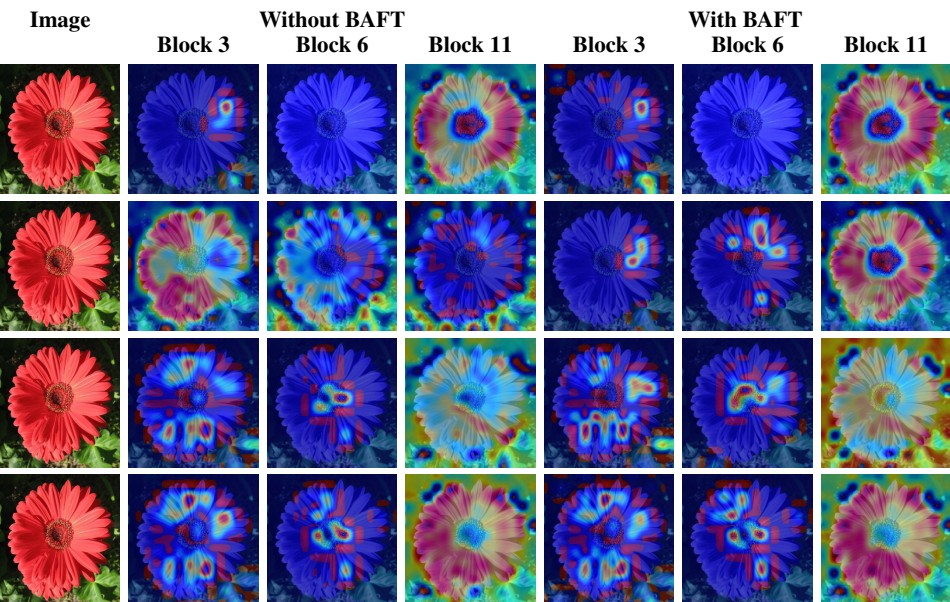

Figure 4: Grad-CAM (Selvaraju et al., 2016) visualizations of AdaptFormer, Bi-AdaptFormer, LoRA, and Bi-LoRA with and without BAFT. BAFT guides fine-tuning using output-space signals, consistently producing sharper attention on petals and stamens.

## 4.1 EXPERIMENTAL SETUP

**Datasets.** To evaluate the effectiveness and robustness of our method, we conduct experiments on two challenging benchmarks: VTAB-1k and few-shot fine-grained visual classification (FGVC). *VTAB-1k* comprises 19 diverse datasets across Natural (7), Specialized (4), and Structured (8) categories, each with 1,000 labeled examples. Following (Jia et al., 2022; Jie et al., 2023), we use the standard 800/200 split for hyperparameter selection, train on the full set, and report average accuracy over three trials, testing generalization across varied visual domains. *Few-shot FGVC* evaluates adaptability in low-data regimes. We consider five fine-grained datasets: FGVC-Aircraft (Maji et al., 2013), Food-101 (Bossard et al., 2014), Oxford Flowers (Nilsback & Zisserman, 2006), Oxford Pets (Parkhi et al., 2012), and Stanford Cars (Krause et al., 2013), using 1, 2, 4, 8, and 16 shots per class. Models are trained on the provided training sets, hyperparameters tuned on validation sets, and average test accuracy is reported over three trials.

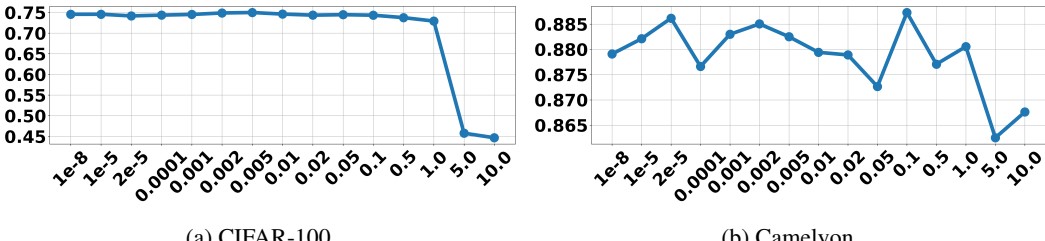

(a) CIFAR-100.              (b) Camelyon.

Figure 5: $\lambda$ sensitivity on CIFAR-100 and Camelyon (VTAB-1k). Optimal $\lambda$ varies by dataset, balancing task supervision and structural alignment; overly strong regularization impairs performance.

**Setups.** All experiments use a ViT-B/16 (Dosovitskiy et al., 2020) backbone pre-trained on supervised ImageNet-21K (Deng et al., 2009). Following (Jia et al., 2022; Jie et al., 2023), we adopt the Adam optimizer with a cosine learning rate schedule and 10-epoch linear warm-up. For VTAB-1k, images are resized to 224×224; for few-shot FGVC, images are resized to 256×256 then center-cropped. Models are fine-tuned for 300 epochs on VTAB-1k and 100 epochs on FGVC, using a batch size of 64. To ensure fair comparisons, we retrain all PEFT baselines using their original hyperparameters. For LoRA and AdaptFormer, the hidden size is set to 8. We then perform a grid search over $\lambda$, following established tuning protocols (Zhai et al., 2019; Jia et al., 2022; Jie et al., 2023), ensuring a robust evaluation of our method's added value. See Appendix for more details.

## 4.2 COMPARISON WITH THE STATE-OF-THE-ART

**Behaviour-aligned insights.** Fig. 4 visualizes Grad-CAM activations for Adapt-Former, Bi-AdaptFormer, LoRA, and Bi-LoRA with and without BAFT. Incorporating BAFT uses output-space signals to guide fine-tuning, consistently producing sharper, more focused attention on discriminative features such as petals and stamens. This demonstrates that BAFT transforms fine-tuning into a guided process, aligning feature learning with the task-relevant output behavior throughout the network (see also Fig. 1).

**Results on VTAB-1k.** We evaluate our method against a wide range of baselines, including full fine-tuning, linear probing, and leading PEFT approaches such as LoRA, AdaptFormer, Bi-LoRA, and Bi-AdaptFormer. Table 1 presents the full results. Across all dataset categories, integrating BAFT consistently improves each PEFT method. Notably, our strongest model, Bi-AdaptFormer + BAFT, achieves an average accuracy of 77.5% across 19 diverse tasks, outperforming all baselines, including full fine-tuning, which reaches only 68.9% while using 145× more trainable parameters (85.8M vs. 0.59M). Other PEFT methods also benefit from BAFT: AdaptFormer improves from 76.9% to 77.3%, LoRA from 76.4% to 76.8%, and Bi-LoRA from 76.7% to 77.1%. These

Table 2: Results on VTAB-1k using ViT-Large pre-trained on ImageNet-21k as the backbone.

| Method | Natural | Specialized | Structured | **Mean** | **#Params(M)** |
|---|---|---|---|---|---|
| AdaptFormer | 83.8 | 86.0 | 61.0 | 76.9 | 0.42 |
| **+ BAFT** | 83.9 | 86.4 | 61.6 | 77.3 | 0.42 |
| LoRA | 83.7 | 85.7 | 61.6 | 77.0 | 0.81 |
| **+ BAFT** | 83.8 | 86.2 | 62.0 | 77.3 | 0.81 |
| Bi-AdaptFormer | 84.0 | 86.2 | 61.4 | 77.2 | 1.60 |
| **+ BAFT** | 84.2 | 86.5 | 62.2 | 77.6 | 1.60 |
| Bi-LoRA | 83.2 | 85.0 | 61.7 | 76.6 | 3.17 |
| **+ BAFT** | 83.4 | 85.4 | 62.1 | 77.0 | 3.17 |

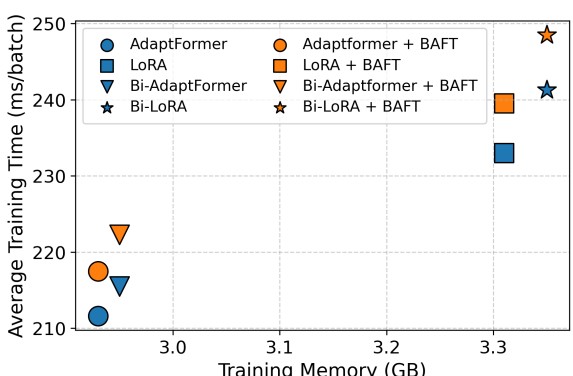

Figure 6: Training efficiency with/without BAFT on NVIDIA A4000 (batch size 64). $x$: GPU memory (GB); $y$: step time (ms). BAFT adds minor overhead.

consistent gains, achieved without adding trainable parameters, highlight the effectiveness and compatibility of our behavior-guided approach. Performance gains are particularly notable in the Structured category, which includes tasks requiring spatial and relational reasoning (*e.g.*, Clevr-Count, KITTI-Dist, and sNORB-Ele). For instance, Bi-AdaptFormer + BAFT improves accuracy on Clevr-Count from 84.9% to 85.1%, KITTI-Dist from 82.1% to 82.3%, and sNORB-Ele from 40.5% to 40.7%. While some improvements appear modest, their consistency across all datasets underscores BAFT's generality, robustness, and task-awareness.

**Results on larger-scale ViT backbone.** Beyond ViT-B, we also evaluate our method on the larger ViT-L backbone, which features a deeper block structure, to test scalability. As shown in Table 2, our method consistently outperforms all selected baselines while maintaining a reasonable parameter budget. These results demonstrate that our approach effectively adapts to models of varying scales.

**Results on few-shot learning.** Figure 3 illustrates few-shot performance across five FGVC datasets, comparing AdaptFormer, AdaptFormer + BAFT, LoRA, and LoRA + BAFT under varying shot

counts. Across all settings, integrating BAFT consistently improves or matches baseline performance. Notably, AdaptFormer + BAFT achieves the highest accuracies on FGVC-Aircraft and Oxford-Pets in higher-shot regimes, while LoRA + BAFT delivers substantial gains in low-shot scenarios, particularly on Food101 and FGVC-Aircraft. The averaged trends show a consistent improvement with BAFT, highlighting its ability to enhance generalization under limited supervision and reinforce task-relevant feature alignment in few-shot adaptation.

**Computational efficiency.** Fig. 6 compares training efficiency across baselines and our method. The $x$-axis shows peak GPU memory usage (GB), and the $y$-axis reports average training time per step (ms). Among baselines, AdaptFormer and Bi-AdaptFormer are more efficient, requiring less memory and runtime than LoRA and Bi-LoRA. Adding BAFT introduces a modest overhead in both memory and runtime, reflecting the cost of the additional regularization. Importantly, this overhead is small, about 0.02 to 0.05 GB in memory and 10 to 15 ms in runtime, demonstrating that BAFT can be integrated into existing PEFT methods with minimal computational cost.

Additional results, evaluations, and visualizations are provided in the Appendix.

### 4.3 Ablation Study

**Impact of** $\lambda$**.** We evaluate the effect of the $\lambda$ hyperparameter through a sensitivity analysis on CIFAR-100 and Camelyon (VTAB-1k), varying $\lambda$ across several orders of magnitude. As shown in Figure 5, both datasets demonstrate strong robustness to a wide range of small $\lambda$ values. CIFAR-100 achieves peak accuracy near $\lambda = 0.002$ and remains stable from $1e$–8 to 0.1. Similarly, Camelyon maintains high accuracy ($\sim$88%) across small to moderate $\lambda$, peaking around 0.1. However, performance sharply declines when $\lambda > 1.0$, indicating that excessive regularization impairs learning. This is es-

Table 3: Impact of BAFT placement in Bi-AdaptFormer. Early placement refers to the first three layers, and late placement to the last three layers. Applying BAFT across all layers achieves the highest accuracy, surpassing both early-only and late-only placements.

|       | Natural | Specialized | Structured | Avg. (%) |
|-------|---------|-------------|------------|----------|
| Early | 82.5    | 86.2        | 63.0       | 77.2     |
| Late  | 82.5    | 86.2        | 63.1       | 77.3     |
| **All** | **82.7** | **86.5**   | **63.2**   | **77.5** |

pecially pronounced in CIFAR-100, where accuracy drops from 74% to below 45% as $\lambda$ increases from 1.0 to 10.0. Camelyon shows a similar but milder trend, likely due to its simpler structure and higher class separability.

**Placement of BAFT.** We investigate the optimal placement of the BAFT module within the Bi-AdaptFormer architecture to assess whether its location influences performance. Specifically, we apply BAFT at three different scopes: *early* layers, *late* layers, and *all* layers of the backbone. Table 3 summarizes the results. Applying BAFT only to the early layers yields a group-wise average accuracy of 77.2%, while applying it to the late layers slightly improves performance to 77.3%. The highest accuracy, 77.5%, is achieved when BAFT is applied across all layers. This trend suggests that although both early and late layers benefit from behavioral alignment, a full-network application provides the most comprehensive and synergistic guidance, leading to the best overall performance.

Further discussion on representation dynamics is provided in Appendix A.7.

## 5 Conclusion

We proposed *a behaviorally guided strategy* for PEFT that aligns intermediate representations with the relational structure inherent in model predictions. This lightweight, parameter-free approach integrates seamlessly into existing PEFT pipelines. Through theoretical insights and empirical results, we demonstrate that this alignment shapes internal representations in a task-aware and semantically meaningful way, addressing a key limitation in current PEFT methods, which *often overlook the dynamics of adaptation*. Our method consistently boosts performance across VTAB-1k and few-shot transfer tasks. By turning prediction structure into actionable training signals, this work *opens a promising direction for fine-tuning*: one that is efficient, behaviorally aligned, and well-suited to data-scarce regimes. In future work, we plan to extend BAFT beyond classification to domains such as detection and segmentation to evaluate the generality and scalability of the approach.

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

# A   APPENDIX

Table 4: VTAB-1k datasets categorized into Natural, Specialized, and Structured groups. Training set sizes are 800 or 1,000 depending on availability.

| Category | Dataset | # Classes | Train | Val | Test |
|---|---|---|---|---|---|
| Natural | CIFAR100 | 100 | | | 10,000 |
| | Caltech101 | 102 | | | 6,084 |
| | DTD | 47 | | | 1,880 |
| | Oxford-Flowers102 | 102 | 800/1,000 | 200 | 6,149 |
| | Oxford-Pets | 37 | | | 3,669 |
| | SVHN | 10 | | | 26,032 |
| | Sun397 | 397 | | | 21,750 |
| Specialized | Patch Camelyon | 2 | | | 32,768 |
| | EuroSAT | 10 | | | 5,400 |
| | Resisc45 | 45 | 800/1,000 | 200 | 6,300 |
| | Retinopathy | 5 | | | 42,670 |
| Structured | Clevr/count | 8 | | | 15,000 |
| | Clevr/distance | 6 | | | 15,000 |
| | DMLab | 6 | | | 22,735 |
| | KITTI-Dist | 4 | 800/1,000 | 200 | 711 |
| | dSprites/location | 16 | | | 73,728 |
| | dSprites/orientation | 16 | | | 73,728 |
| | SmallNORB/azimuth | 18 | | | 12,150 |
| | SmallNORB/elevation | 18 | | | 12,150 |

Table 5: Few-shot datasets used for evaluation. Training size varies (e.g., 1/2/4/8/16 per class), with fixed validation and test sets.

| Dataset | # Classes | Train | Val | Test |
|---|---|---|---|---|
| Food-101 | 101 | | 20,200 | 30,300 |
| Stanford Cars | 196 | | 1,635 | 8,041 |
| Oxford-Flowers102 | 102 | 1/2/4/8/16 per class | 1,633 | 2,463 |
| FGVC-Aircraft | 100 | | 3,333 | 3,333 |
| Oxford-Pets | 37 | | 736 | 3,669 |

## A.1   LLM USAGE DECLARATION

We disclose the use of Large Language Models (LLMs) as general-purpose assistive tools during the preparation of this manuscript. LLMs were used only for minor tasks such as grammar and style improvement, code verification, and formatting suggestions. No scientific ideas, analyses, experimental designs, or conclusions were generated by LLMs. All core research, methodology, experiments, and results were performed and fully verified by the authors.

The authors take full responsibility for all content presented in this paper, including text or code suggestions that were refined with the assistance of LLMs. No content generated by LLMs was treated as original scientific work, and all references and claims have been independently verified. LLMs did not contribute in a manner that would qualify them for authorship.

## A.2   INTERACTION WITH PEFT

Unlike contrastive or distillation-based objectives that rely on fixed anchors or teacher networks, our alignment mechanism uses the model's own evolving predictions to derive a dynamic, task-aware supervisory signal. This gives rise to a closed-loop interaction between output-space structure and representation learning, which we formalize as follows.

Let $\boldsymbol{Z} \in \mathbb{R}^{B \times d}$ and $\boldsymbol{P} \in \mathbb{R}^{B \times C}$ denote $\ell_2$-normalized intermediate feature and prediction matrices, respectively. The alignment loss induces gradient signals of the form:

$$\frac{\partial \mathcal{L}_{\text{BAFT}}}{\partial \boldsymbol{Z}} = 4(\boldsymbol{Z}\boldsymbol{Z}^\top - \boldsymbol{P}\boldsymbol{P}^\top)\boldsymbol{Z}. \tag{7}$$

This term reveals a rich structure: the update to $\boldsymbol{Z}$ depends not only on its current geometry, but on how it diverges from the pairwise relational structure implied by $\boldsymbol{P}$. Crucially, $\boldsymbol{P}$ itself is produced by a forward pass through the model, so the alignment loss introduces an implicit coupling between feature learning and prediction dynamics.

We can characterize the learning process as a form of self-consistent geometric shaping: as $\boldsymbol{P}$ becomes more structured (*e.g.*, grouping same-class examples), $\mathcal{L}_{\text{BAFT}}$ encourages $\boldsymbol{Z}$ to mirror this geometry, which in turn sharpens $\boldsymbol{P}$ through more discriminative features. Formally, this feedback loop induces a time-dependent dynamical system:

$$\frac{d\boldsymbol{Z}}{dt} = -\nabla_{\boldsymbol{Z}} \left( \mathcal{L}_{\text{Sup}} + \lambda \mathcal{L}_{\text{BAFT}} \right), \tag{8}$$

where the trajectory of $\boldsymbol{Z}(t)$ is modulated not only by the supervised signal, but by a structure-aware regularizer that tracks and reinforces the model's own evolving belief space. This self-reinforcing mechanism contrasts with static priors or handcrafted losses: it adapts *per batch*, *per step*, and *per model state*. Importantly, it encourages the formation of semantically coherent and linearly separable manifolds in feature space, a property known to benefit generalization and transfer, especially under limited data or constrained optimization settings such as PEFT.

PEFT methods, such as LoRA and AdaptFormer, operate by restricting model updates to a low-dimensional subspace of the full parameter space. Let $\boldsymbol{\theta} = \boldsymbol{\theta}_0 + \Delta_{\boldsymbol{\phi}}$, where $\boldsymbol{\theta}_0$ is frozen and $\boldsymbol{\phi}$ are the tunable adapter parameters. Then the model output is:

$$f_{\boldsymbol{\theta}}(x) = f_{\boldsymbol{\theta}_0 + \Delta_{\boldsymbol{\phi}}}(\boldsymbol{x}). \tag{9}$$

The optimization is restricted to $\boldsymbol{\phi}$, limiting the expressiveness of updates. However, this restriction does not preclude which directions in representation space are preferred, only how those directions are reached.

Our alignment loss serves as a high-level, geometry-aware guide within this constrained space. Even though $\Delta_{\boldsymbol{\phi}}$ lies in a low-rank subspace, the gradient signal from $\mathcal{L}_{\text{BAFT}}$ acts to bias those updates toward directions that reduce inconsistency between internal and output-space structure. This gives the optimization trajectory a relational inductive bias. In other words, PEFT defines where the model can move (low-rank subspaces), but not why it should move in any particular direction. Our method complements this by injecting semantic structure into the optimization, not through external supervision, but through emergent internal consistency constraints. Moreover, this interaction is synergistic: PEFT ensures training remains efficient, while alignment loss prevents degenerate adaptation (*e.g.*, memorization or collapse), especially in low-data regimes where task-relevant geometry is hard to extract from labels alone.

### A.3 DETAILED DATASET STATISTICS

We provide detailed information about the datasets used in this paper, including the number of classes and the sizes of the training, validation and test sets in Table 4 (VTAB-1k) and Table 5(Few-shot learning). The VTAB-1k datasets consists of three categories: Natural, Specialized and Structured tasks. The Natural category includes datasets such as CIFAR-100, Caltech101, DTD, Flowers102, Pets, SVHN, and Sun397; Specialized category includes datasets such as Patch Camelyon, EuroSAT, Resisc45, and Diabetic-Retinopathy and the Structured category also includes datasets such as Clevr/count, Clevr/distance, DMLab, KITTI/distance, dSprites/location, dSprites/orientation, SmallNORB/azimuth, and SmallNORB/elevation. For the the few-shot learning, the datasets consists of fine-grained visual classification(FGVC) including FGVC-Aircraft, Food101, Oxford-Flowers102,Oxford-Pets and Stanford Cars.

### A.4 PROOF OF THEOREM 1

*Proof.* Let $\boldsymbol{K}_Z := \boldsymbol{Z}\boldsymbol{Z}^\top \in \mathbb{R}^{B \times B}$ and $\boldsymbol{K}_P := \boldsymbol{P}\boldsymbol{P}^\top \in \mathbb{R}^{B \times B}$ denote the similarity kernels in feature space and prediction space, respectively. Both matrices are symmetric positive semi-definite

Table 6: Experiment configurations for VTAB-1k and few-shot learning experiments.

| Dataset | optimizer | batch size | learning rate | weight decay | # epochs | lr decay | # warm-up epochs |
|---|---|---|---|---|---|---|---|
| VTAB-1K | AdamW | 64 | 1e-3 | 1e-4 | 300 | cosine | 10 |
| Few-shot learning | AdamW | 64 | 5e-3 | 1e-4 | 100 | cosine | 10 |

Table 7: Best $\lambda$ for the VTAB-1k

| Method | Natural | | | | | | | Specialized | | | | Structured | | | | | | | |
|---|---|---|---|---|---|---|---|---|---|---|---|---|---|---|---|---|---|---|---|
| | Cifar100 | Caltech101 | DTD | Flowers102 | Pets | SVHN | Sun397 | Camelyon | EuroSAT | Resisc45 | Retinopathy | Clevr-Count | Clevr-Dist | DMLab | KITTI-Dist | dSpr-Loc | dSpr-Ori | sNORB-Azim | sNORB-Ele |
| AdaptFormer + **BAFT** | 0.01 | 2e-3 | 0.01 | 0.02 | 1e-8 | 0.01 | 2e-3 | 0.05 | 0.1 | 0.01 | 2e-3 | 0.01 | 0.5 | 0.01 | 0.01 | 0.1 | 2e-3 | 0.1 | 0.05 |
| LoRA + **BAFT** | 1e-3 | 1e-4 | 5e-3 | 1e-5 | 5e-3 | 0.01 | 1e-5 | 0.1 | 2e-3 | 1e-3 | 2e-3 | 0.01 | 1e-3 | 0.01 | 0.01 | 1e-3 | 0.5 | 0.01 | 0.1 |
| Bi-AdaptFormer + **BAFT** | 2e-3 | 2e-3 | 0.01 | 0.01 | 2e-3 | 0.1 | 1e-5 | 0.01 | 0.01 | 0.1 | 0.01 | 0.02 | 0.1 | 0.01 | 0.02 | 0.01 | 0.5 | 0.01 | 0.1 |
| Bi-LoRA + **BAFT** | 0.01 | 1e-8 | 0.01 | 1e-8 | 0.1 | 5e-3 | 1e-5 | 0.2 | 0.1 | 0.05 | 0.1 | 0.01 | 0.05 | 0.01 | 0.01 | 0.01 | 0.2 | 1e-3 | 0.01 |

(SPSD), with rank at most $\min(B, d)$ and $\min(B, C)$. The Frobenius alignment loss is given by:

$$\mathcal{L}_{\text{BAFT}} = \|\boldsymbol{K}_Z - \boldsymbol{K}_P\|_F^2 = \text{Tr}(\boldsymbol{K}_Z^2) + \text{Tr}(\boldsymbol{K}_P^2) - 2\text{Tr}(\boldsymbol{K}_Z \boldsymbol{K}_P). \quad (10)$$

Since $\boldsymbol{K}_Z$ and $\boldsymbol{K}_P$ are SPSD, they admit spectral decompositions:

$$\boldsymbol{K}_Z = \boldsymbol{U}\boldsymbol{\Lambda}\boldsymbol{U}^\top, \quad \boldsymbol{K}_P = \boldsymbol{V}\boldsymbol{\Sigma}\boldsymbol{V}^\top, \quad (11)$$

where $\boldsymbol{U}, \boldsymbol{V} \in \mathbb{R}^{B \times B}$ are orthonormal matrices and $\boldsymbol{\Lambda}, \boldsymbol{\Sigma}$ are diagonal matrices of non-negative eigenvalues. Using the cyclic trace identity, we have:

$$\text{Tr}(\boldsymbol{K}_Z \boldsymbol{K}_P) = \text{Tr}(\boldsymbol{U}\boldsymbol{\Lambda}\boldsymbol{U}^\top \boldsymbol{V}\boldsymbol{\Sigma}\boldsymbol{V}^\top) = \text{Tr}(\boldsymbol{\Lambda}\boldsymbol{U}^\top \boldsymbol{V}\boldsymbol{\Sigma}\boldsymbol{V}^\top \boldsymbol{U}). \quad (12)$$

Let $\boldsymbol{Q} = \boldsymbol{U}^\top \boldsymbol{V}$ be the orthogonal matrix capturing the relative orientation of the eigenspaces. Then:

$$\text{Tr}(\boldsymbol{K}_Z \boldsymbol{K}_P) = \text{Tr}(\boldsymbol{\Lambda}\boldsymbol{Q}\boldsymbol{\Sigma}\boldsymbol{Q}^\top) = \sum_{i=1}^{B} \sum_{j=1}^{B} \lambda_i \sigma_j \boldsymbol{Q}_{ij}^2, \quad (13)$$

which represents an inner product between the eigenspaces of $\boldsymbol{K}_Z$ and $\boldsymbol{K}_P$, weighted by their eigenvalues. By the von Neumann trace inequality (Von Neumann, 1937):

$$\text{Tr}(\boldsymbol{K}_Z \boldsymbol{K}_P) \leq \sum_{i=1}^{B} \lambda_i \sigma_i, \quad (14)$$

with equality if and only if the eigenvectors of $\boldsymbol{K}_Z$ and $\boldsymbol{K}_P$ are aligned. Therefore, minimizing $\mathcal{L}_{\text{BAFT}}$ is equivalent to maximizing $\text{Tr}(\boldsymbol{K}_Z \boldsymbol{K}_P)$, under fixed Frobenius norms $\|\boldsymbol{K}_Z\|_F^2$ and $\|\boldsymbol{K}_P\|_F^2$. The loss reaches its minimum when $\boldsymbol{K}_Z = \boldsymbol{K}_P$, that is, when their eigenvalues match and their eigenspaces are aligned. $\square$

## A.5 IMPLEMENTATION DETAILS

In our experiments, we choose ViT-B/16 trained on ImageNet-21K as our backbone. For VTAB-1k, we follow (Jia et al., 2022) to resize the images to $224 \times 224$. Different from VTAB-1k, we follow (Zhang et al., 2024), using random augmentations during training, for validation/test samples, we resize them to $256 \times 256$, crop them to $224 \times 224$ at the center, and then normalize them with ImageNet's mean and standard deviation. Table 6 shows our experiment configurations.

## A.6 HYPERPARAMETER TUNING

For each dataset, we conduct a hyperparameter search on each dataset to find the best $\lambda$ to optimize performance. We follow the strategies from previous work (Jia et al., 2022; Zhai et al., 2019). We apply grid search on $\lambda \in \{1e-8, 1e-5, 2e-5, 1e-4, 1e-3, 2e-3, 5e-3, 0.01, 0.02, 0.05, 0.1, 0.5, 1.0, 5.0, 10.0\}$. Table 7 presents the best $\lambda$ for the VTAB-1k dataset.

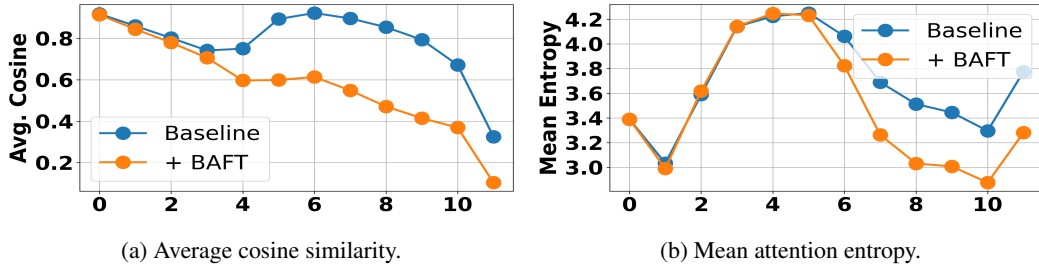

(a) Average cosine similarity.

(b) Mean attention entropy.

Figure 7: Representation dynamics with BAFT. Our method suppresses feature collapse and sharpens attention, promoting task-aligned feature diversity and focused attention distributions.

### A.7 DISCUSSION ON REPRESENTATION DYNAMICS

To better understand the representational impact of BAFT, we analyze how it shapes the internal dynamics of the backbone. We focus on two phenomena common in transformer-based models: *feature collapse suppression* and *attention entropy reduction*. The analysis is conducted using Bi-AdaptFormer trained on CIFAR-100 from VTAB-1k.

**Feature collapse suppression.** Feature collapse occurs when token representations across different samples become overly similar, which limits the model's expressiveness and discriminative power. This effect is especially pronounced in deeper layers due to oversmoothing, as discussed in prior works (Gong et al., 2021; Diko et al., 2024; Wang et al., 2022). To quantify it, we compute the average pairwise cosine similarity among $N$ $\ell_2$-normalized token features $\boldsymbol{x}_1, \ldots, \boldsymbol{x}_N$:

$$\text{AvgCosine} = \frac{2}{N(N-1)} \sum_{1 \le i < j \le N} \boldsymbol{x}_i^\top \boldsymbol{x}_j. \tag{15}$$

As shown in Figure 7a, models trained without BAFT exhibit increasing similarity in deeper layers, indicating a collapse in representation diversity. In contrast, BAFT maintains lower similarity across layers. For example, in the final layer, the average similarity remains below $0.1$ with BAFT, compared to over $0.3$ without it. These results suggest that BAFT promotes feature diversity by softly discouraging semantic redundancy, thereby preventing collapse and enabling richer task-specific representations.

**Attention entropy reduction.** Beyond feature geometry, we examine attention entropy (Zhai et al., 2023) to evaluate how BAFT influences the model's focus and selectivity. A lower entropy reflects more confident and concentrated attention, which is especially beneficial in low-data settings. This sharpened focus helps reinforce informative patterns while suppressing noise, thereby reducing overfitting.

Let $\boldsymbol{A}^{(l,h)} \in \mathbb{R}^{N \times N}$ be the attention matrix at layer $l \in 1, \ldots, L$ and head $h \in 1, \ldots, H$, where each row $\boldsymbol{A}_i^{(l,h)}$ represents a probability distribution over input tokens. The entropy for head $(l, h)$ is:

$$\mathcal{H}^{(l,h)} = -\frac{1}{N} \sum_{i=1}^{N} \sum_{j=1}^{N} \boldsymbol{A}_{i,j}^{(l,h)} \log \boldsymbol{A}_{i,j}^{(l,h)}. \tag{16}$$

We compute the mean attention entropy across all layers and heads as:

$$\text{MeanEntropy} = \frac{1}{L \cdot H} \sum_{l=1}^{L} \sum_{h=1}^{H} \mathcal{H}^{(l,h)}. \tag{17}$$

As shown in Figure 7b, models with BAFT exhibit noticeably lower entropy from layer 6 onward, indicating more focused and selective attention. The largest reductions occur in the final layers, suggesting that BAFT enhances attention sharpness and helps the model prioritize informative regions during adaptation.

Table 8: Results on VTAB-1k using Bi-AdaptFormer with different training epochs.

| Epoch | Method | Natural | Specialized | Structured | Average |
|-------|--------|---------|-------------|------------|---------|
| 100 | Bi-AdaptFormer | 82.2 | 85.2 | 62.6 | 76.7 |
| 100 | + BAFT | 82.4 | 86.2 | 62.4 | 77.0 |
| 200 | Bi-AdaptFormer | 82.4 | 85.9 | 62.7 | 77.0 |
| 200 | + BAFT | 82.5 | 86.1 | 62.8 | 77.1 |
| 300 | Bi-AdaptFormer | 82.5 | 85.6 | 62.9 | 77.0 |
| 300 | + BAFT | 82.7 | 86.5 | 63.2 | 77.5 |

Table 9: Results on VTAB-1k using Swin-Base pretrained on ImageNet-21k as the backbone.

| Method | Natural | Specialized | Structured | Average |
|--------|---------|-------------|------------|---------|
| AdaptFormer | 82.5 | 87.0 | 60.7 | 76.7 |
| + BAFT | 82.6 | 87.1 | 60.8 | 76.8 |
| LoRA | 82.3 | 86.7 | 61.9 | 77.0 |
| + BAFT | 82.5 | 87.0 | 62.2 | 77.2 |
| Bi-AdaptFormer | 82.3 | 87.2 | 61.1 | 76.9 |
| + BAFT | 82.4 | 87.4 | 61.4 | 77.1 |
| Bi-LoRA | 82.1 | 87.2 | 61.5 | 76.9 |
| + BAFT | 82.3 | 87.3 | 61.9 | 77.2 |

> These analyses show that BAFT shapes both the geometry and attention dynamics of the network. By encouraging behaviorally aligned similarity in feature space and promoting focused attention distributions, BAFT produces more discriminative and robust representations. This is particularly beneficial in low-data adaptation scenarios, where overfitting and feature degeneration are common challenges.

## A.8 ADDITIONAL EXPERIMENTS

In this section, we provide additional experiments of our baseline models with BAFT. Specifically, using the Bi-AdaptFormer model, we check the impact of different epochs with the VTAB-1k dataset. Additionally, we also experiment with hierarchical ViT backbones(Swin-Transformer).

**Experiments on VTAB-1k with different epochs.** All main models were trained for 300 epochs across the VTAB-1K tasks, ensuring consistency and fair comparison of performance across PEFT methods such as Adaptformer, LoRA, Bi-Adaptformer, and Bi-LoRA. To further explore training dynamics and sensitivity to epoch count, we conducted additional experiments varying the number of epochs and reported these results separately in Table 8. This provides insight into model stability and convergence behavior under different training durations.

**Experiments on VTAB-1k with Swin Transformer.** Table 9 displays the experimental results on VTAB-1k using Swin-Base pretrained on ImageNet-21k as the backbone.

## A.9 LIMITATIONS

While BAFT offers a lightweight, parameter-free approach for behaviorally guided adaptation, several limitations should be noted. First, in our current implementation, the prediction vectors are not detached from the computational graph, so the auxiliary BAFT loss can directly influence the prediction logits. In practice, this did not destabilize training, but introducing a stop-gradient or an EMA-teacher variant could provide a cleaner separation between feature alignment and prediction dynamics, potentially improving stability and interpretability.

Second, our main experiments apply BAFT uniformly across all layers with equal weighting. This simple approach may not fully leverage the differing semantic contributions of early versus late layers. Future work could explore adaptive per-layer weighting schemes or dynamic strategies that

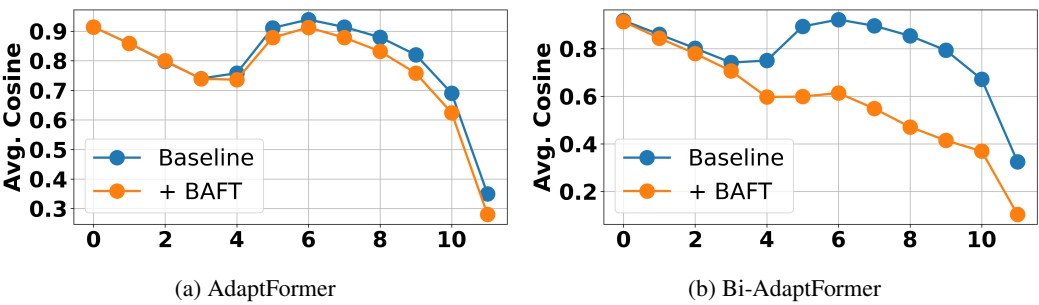

(a) AdaptFormer  (b) Bi-AdaptFormer

Figure 8: Impact of BAFT in terms of feature collapse suppression on CIFAR-100 (VTAB-1k).

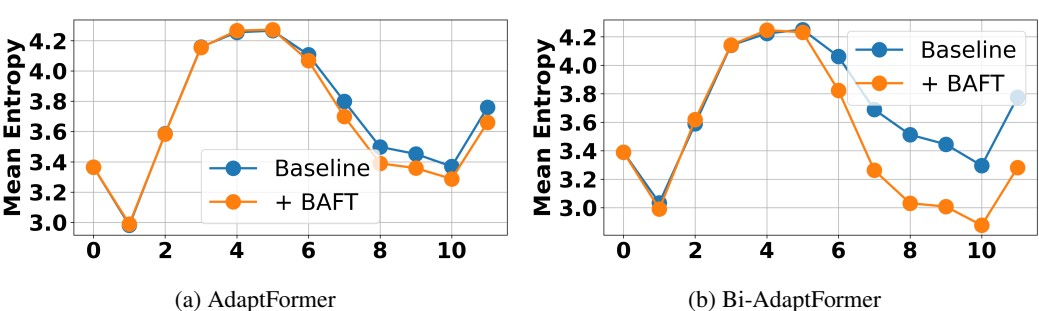

(a) AdaptFormer  (b) Bi-AdaptFormer

Figure 9: Impact of BAFT in terms of attention entropy reduction on CIFAR-100 (VTAB-1k).

emphasize layers most relevant for the task, which may further enhance feature alignment and task performance.

Finally, while BAFT is parameter-free and efficient, its effectiveness may vary across architectures or tasks with highly complex or heterogeneous output structures. Comprehensive evaluations on diverse architectures and tasks would help clarify the generality and limitations of the method.

## A.10 IMPACT OF BAFT

**Feature collapse suppression.** Figure 8 shows the impact of BAFT in terms of suppressing feature collapse, measuring the average pairwise cosine similarity in depth. The figure plots the average pairwise cosine similarity on the CIFAR-100 dataset. These results suggest that BAFT promotes feature diversity by softly discouraging semantic redundancy, thereby preventing collapse and enabling richer task-specific representations.

**Attention entropy reduction.** Figure 9 shows the impact of BAFT in terms of reducing attention entropy, by measuring the mean attention entropy in depth. The figure plots the mean attention entropy on the CIFAR-100(VTAB-1k) dataset. Models with BAFT exhibit noticeably lower entropy from layer 6 onward, indicating more focused and selective attention. The largest reductions occur in the final layers, suggesting that BAFT enhances attention sharpness and helps the model prioritize informative regions during adaptation.

## A.11 MORE VISUALIZATIONS

Figs. 10, 11, 12, 13 show Grad-CAM visualizations of all of baseline models, comparing with and without BAFT. These results show BAFT's robustness and effectiveness across several datasets of VTAB-1k.

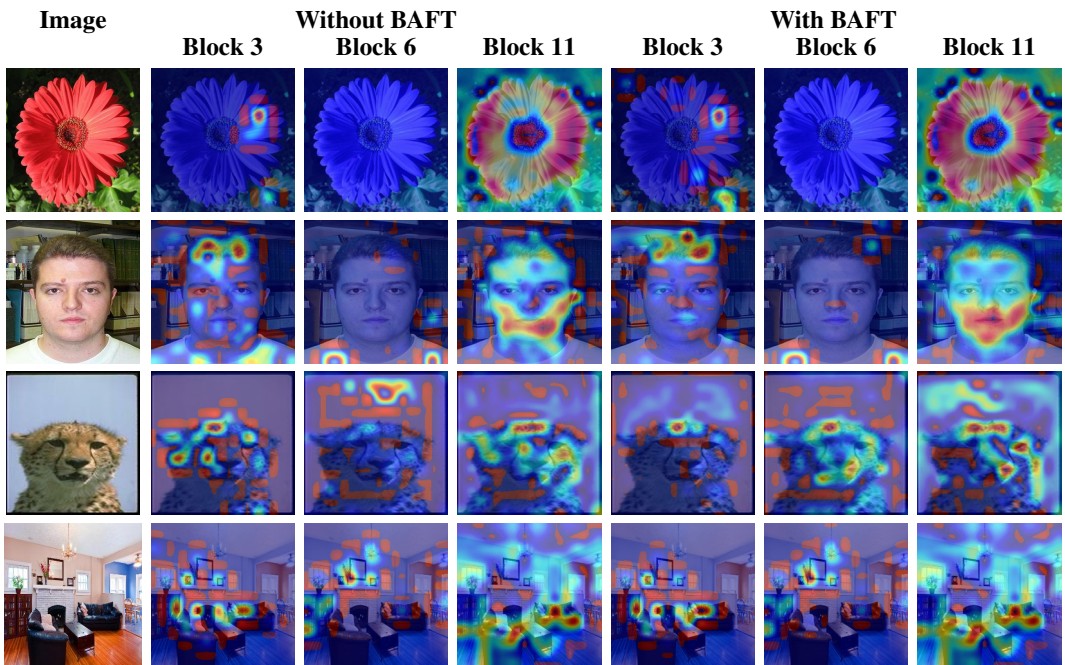

Figure 10: Grad-CAM (Selvaraju et al., 2016) visualizations of the 3rd, 6th and final block of AdaptFormer fine-tuned with and without BAFT on VTAB-1k datasets.

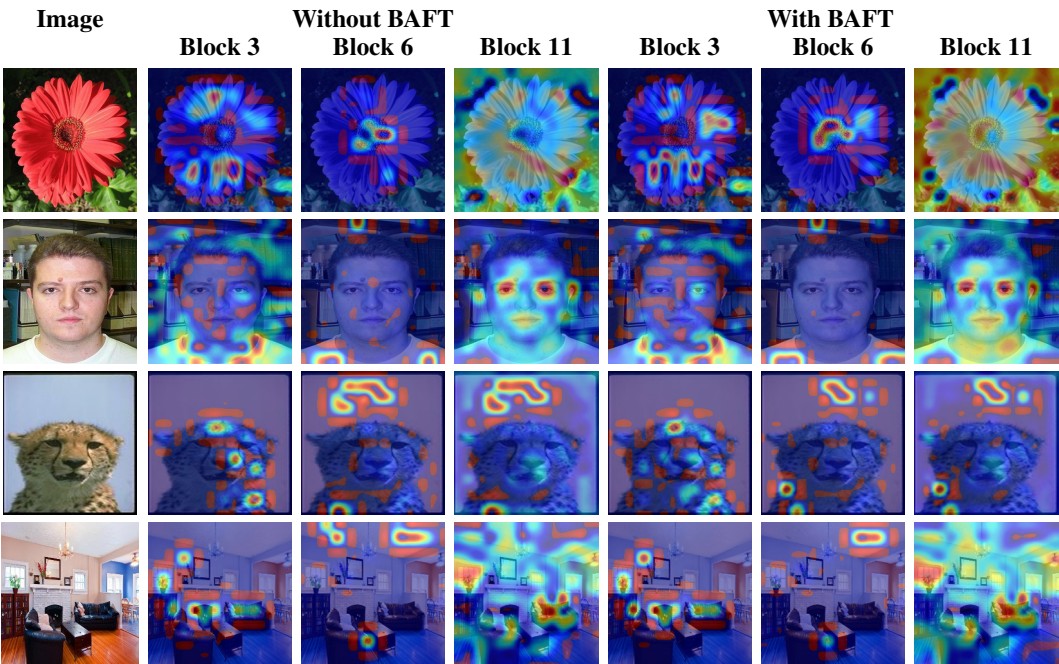

Figure 11: Grad-CAM (Selvaraju et al., 2016) visualizations of the 3rd, 6th and final block of LoRA fine-tuned with and without BAFT on VTAB-1k datasets.

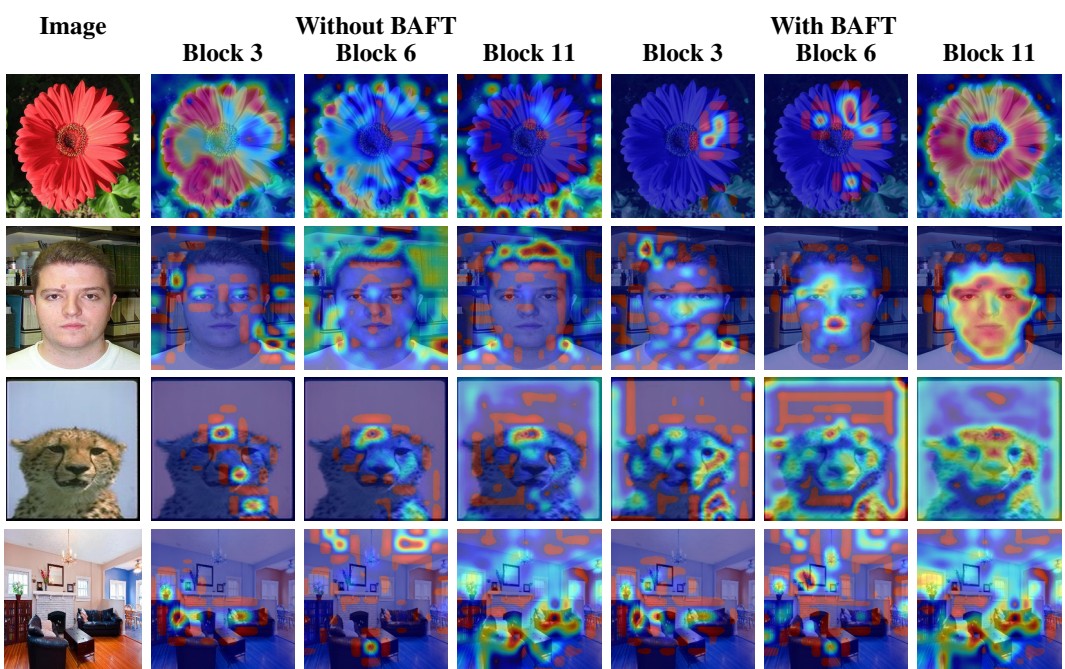

Figure 12: Grad-CAM (Selvaraju et al., 2016) visualizations of the 3rd, 6th and final block of Bi-AdaptFormer fine-tuned with and without BAFT on VTAB-1k datasets.

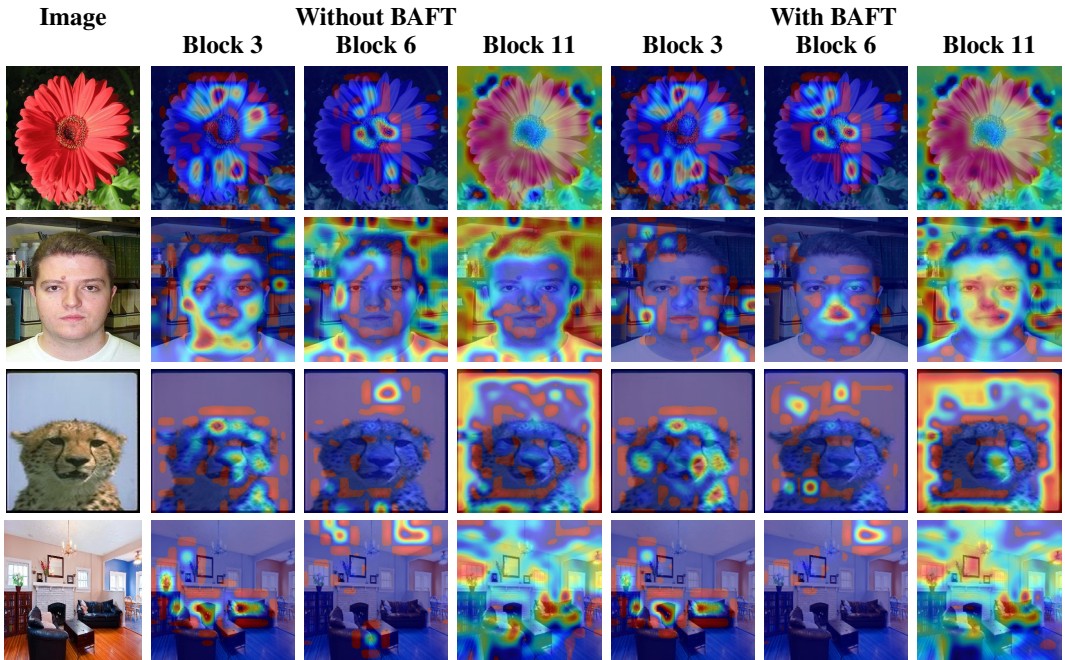

Figure 13: Grad-CAM (Selvaraju et al., 2016) visualizations of the 3rd, 6th and final block of Bi-LoRA fine-tuned with and without BAFT on VTAB-1k datasets.

