# OpenReview forum: "Efficient Fine-Tuning via Behavior-Guided Spectral Alignment"
_ICLR.cc/2026/Conference — Submitted to ICLR 2026_

### Official Review · Reviewer_rM6W · 2025-10-17

**Soundness:** 3
**Presentation:** 3
**Contribution:** 3
**Rating:** 6
**Confidence:** 4

**Summary:**

The paper proposes a parameter-efficient fine-tuning approach that creatively guides via the supervision between feature-level output and model output. Results show that without adding additional parameters, this attempt can increase the performance. The method itself is simple and effective.

**Strengths:**

1. The paper is easy to follow, with the observation that feature-level and output-level supervision can be a useful "self-improvement" on downstreams.

2. The experiments are somewhat comprehensive, while more methods (see weakness) should be included for completeness.

3. The discussions on eigenvalues are sound, and they support the reason for using the behavior-guided alignment. However, the motivation for using it is still unclear to me.

**Weaknesses:**

1. Why do the authors think the distance measurement is significant/important? For a design perspective, it is unnatural to think that the distance across different layers (prediction space and feature space) is important and should align with each other, as the layer-wise outputs may specialize in different levels of pattern recognition. The motivation for doing this is not clear to me.

2. More baselines can be reported in the main results. The latest method included in this study is from 2024. Other PEFT approaches [1-3] can be included for completeness.

3. The feature-level supervision idea is very similar to some knowledge distillation approaches, which supervise pair-wise signals between teacher and student. Do you think it is reasonable to add more discussions on knowledge distillation supervision? I notice that the authors use the word "teacher-free" to represent their supervision signal. Thus, it makes sense to me to add additional discussions [4-6].


[1] Visual Fourier Prompt Tuning

[2] Visual Variational Autoencoder Prompt Tuning

[3] Visual instance-aware prompt tuning

[4] Cross-layer distillation with semantic calibration

[5] Ad-kd: Attribution-driven knowledge distillation for language model compression

[6] AMD: Automatic Multi-step Distillation of Large-scale Vision Models

**Questions:**

Format problem (does not affect my ratings): Sec. 3 Method title is solely on page 3.

---

> ### Author Response · Authors · 2025-11-21
>
> We thank the reviewer for the constructive feedback and for highlighting the paper’s clarity, the usefulness of feature–output relational supervision, and the comprehensiveness of the evaluation.
>
> Below we address all questions and weaknesses.
>
> **1. Why is aligning distances across prediction space and feature space important? Layers may specialize differently; the motivation is unclear.**
>
> BAFT does not enforce layer-level feature matching; instead, it aligns relational structure.
>
> - PEFT often induces misalignment between prediction behavior and feature geometry due to *restricted update capacity*, leading to fragmented attention, inconsistent gradients, and unstable adaptation (observed in Figs. 1, 4, and Appendix A.11).
>
> - Predictions encode the model’s highest-level semantics. When prediction similarity and feature similarity disagree, the internal geometry becomes incoherent.
>
> - The spectral analysis shows that BAFT aligns the eigenspaces of the feature kernel and prediction kernel, enforcing consistent low-frequency structure while preserving specialization of intermediate layers. *BAFT thus regularizes global geometry without collapsing representational diversity.*
>
> When $\lambda = 0$ (Eq. (3) in the main paper), the loss reduces to the standard PEFT objective, indicating that **BAFT provides a more general formulation that augments the original loss with an additional regularization term**.
>
> We have clarified this motivation in the revision.
>
> **2. More baselines should be included for completeness.**
>
> We agree. To improve completeness, we extended our comparison to include recent PEFT approaches such as Visual Fourier Prompt Tuning (VFT), Visual Variational Autoencoder Prompt Tuning (V2PT), and Visual Instance-Aware Prompt Tuning (VIAPT) [1–3]. These methods represent prompt-based PEFT strategies and are complementary to adapter-based or low-rank PEFT.
>
> Since BAFT introduces no architectural changes and no additional parameters, it can be applied directly on top of these methods. The table below summarizes the performance.
>
> | Method   |   Natural  | Specialized | Structured| Avg |
> |-|-|-|-|-|
> | VFT [1]   | 81.4 | 84.9 | 60.2 | 75.5 |
> | **+ BAFT** | **82.1** | **85.4** | **60.5** | **76.0**|
> | V2PT [2]   | 80.0 | 80.5 | 60.0| 73.5 |
> | **+ BAFT** | **81.5** | **80.9** | **61.2** | **74.5**|
> | VIAPT [3]   | 82.6 | 85.2 | 61.3| 76.4 |
> | **+ BAFT** | **83.5** | **85.9** | **62.0** | **77.1**|
>
> **3. Feature-level supervision resembles knowledge distillation; more discussion is needed.**
>
> We appreciate this suggestion.
>
> BAFT is related in spirit to distillation but differs fundamentally:
>
> - Classical KD and cross-layer KD require a separate teacher model; BAFT is *entirely teacher-free and derives relational targets from the model’s own predictions*.
>
> - KD aligns features or logits directly, whereas BAFT aligns *pairwise relational kernels* ($S_z$ and $S_p$).
>
> - KD provides static supervision; *BAFT provides dynamic, self-evolving and structure-aware guidance*.
>
> - KD typically requires additional forward passes or memory sources, while BAFT introduces zero overhead and no architectural change.
>
> We have expanded discussion referencing [4-6] in the camera-ready.

---

> > ### Author Response · Authors · 2025-11-21
> >
> > **4. Motivation for using behavior-guided alignment remains unclear.**
> >
> > Behavior-guided alignment is motivated by a core limitation of PEFT: behavioral misalignment.
> >
> > Because **PEFT methods update only a tiny subset of parameters, they frequently suffer from mismatches between representations and predictions, degraded attention patterns, and noisy or unstable gradients**. These issues are especially pronounced in **low-data regimes**, where *small perturbations can disproportionately distort the learned geometry*.
> >
> > BAFT addresses this failure mode by introducing a principled, parameter-free geometric prior that encourages coherent relational structure in both feature and prediction spaces. Rather than approximating an evaluation metric, BAFT regularizes the geometry of the model’s behavior.
> >
> > From a spectral perspective, BAFT suppresses high-frequency distortions in the relational graphs induced by PEFT updates, effectively **stabilizing the optimization landscape and improving sample efficiency**. This provides a theoretical explanation for *why aligning pairwise structure leads to more reliable adaptation*, even when the *model capacity is heavily constrained*.
> >
> > To further demonstrate the generality of this mechanism beyond vision models, we extend our study to the GLUE benchmark using DeBERTaV3-base.
> >
> > As shown below, BAFT consistently improves both P-Adapter and LoRA across all tasks, reinforcing that behavior-guided alignment helps PEFT methods maintain coherent representations across modalities and architectures.
> >
> > | Method        | #Param | CoLA  | STS-B | RTE   | MRPC  | SST-2 | QNLI  | Avg   |
> > |-|-|-|-|-|-|-|-|-|
> > | P-Adapter     | 1.18M  | 68.57 | 91.45 | 85.34 | 88.90 | 95.53 | 94.19 | 87.33 |
> > | **+ BAFT**        | 1.18M  | **69.10** | **92.58** | **85.65** | **88.98** | **96.05** | **94.23** | **87.77** |
> > | LoRA (r=8)    | 1.33M  | 67.96 | 91.63 | 84.65 | 90.23 | 95.36 | 94.01 | 87.31 |
> > | **+ BAFT**        | 1.33M  | **68.30** | **91.95** | **84.73** | **92.56** | **95.98** | **95.02** | **88.09** |
> >
> > We will strengthen this motivational paragraph.
> >
> > **5. Formatting issue (Section title isolated on page).**
> >
> > We have corrected this LaTeX pagination artifact.
> >
> > We thank the reviewer for the positive evaluation of the soundness, presentation, and contribution.
> >
> > We believe the added clarifications, additional baselines, and extended discussion on distillation will fully address the reviewer’s concerns.
> >
> > References:
> >
> > [1] Visual Fourier Prompt Tuning
> >
> > [2] Visual Variational Autoencoder Prompt Tuning
> >
> > [3] Visual instance-aware prompt tuning
> >
> > [4] Cross-layer distillation with semantic calibration
> >
> > [5] Ad-kd: Attribution-driven knowledge distillation for language model compression
> >
> > [6] AMD: Automatic Multi-step Distillation of Large-scale Vision Models

---

### Official Review · Reviewer_PS61 · 2025-10-23

**Soundness:** 2
**Presentation:** 3
**Contribution:** 2
**Rating:** 2
**Confidence:** 3

**Summary:**

This paper introduces Behavior-Aligned Fine-Tuning (BAFT), a parameter-free, teacher-free add-on for Parameter-Efficient Fine-Tuning (PEFT) methods. Instead of adding new trainable parameters, BAFT aligns intermediate feature representations with the relational structure of the model’s own predictions.

**Strengths:**

1. The idea of this work is interesting -- further exploiting existing training signals as behavioral guidance.
2. The evaluation is thorough.
3. BAFT achieves consistently better results compared to PEFT baselines.

**Weaknesses:**

1. BAFT only seem to be effective for Bi-AdaptFormer as Figure 4 shows, contradicting with "models trained with BAFT consistently produce
 more focused and semantically meaningful activation maps" in introduction.
2. The authors seem to claim that the lack of behavioral oversight is a key limitation of PEFT methods, however, it is not well-supported by experiments, as the improvements with BAFT seem minor.

**Questions:**

1. It's commonly acknowledged that the evaluation metric is better aligned with the loss, so the evaluation metric gets directly optimized. However, L_BAFT is included that does not directly reflect the evaluation metric, why is it not a disturbing term? Could you provide discussions why this approach is better than directly optimizing, e.g., cross-entropy?

---

> ### Author Response · Authors · 2025-11-21
>
> We thank the reviewer for the constructive comments and for recognizing the novelty of using behavioral guidance, the thorough evaluation, and the consistent improvements across PEFT methods.
>
> Below we address all concerns.
>
> **1. BAFT only seems effective for Bi-AdaptFormer (Fig.~4), contradicting the claim that BAFT consistently improves activation maps.**
>
> We thank the reviewer for pointing this out and would like to clarify a misunderstanding.
>
> Figure 4 does not illustrate only Bi-AdaptFormer; rather, it shows blockwise Grad-CAM visualizations across multiple intermediate layers (Blocks 3, 6, 11) for several input images. Across all rows / layers, BAFT produces more spatially concentrated, object-centric, and semantically coherent activation patterns compared to training without BAFT.
>
> Specifically: (i) Without BAFT, activation maps are often diffuse, scattered, or dominated by background regions. (ii) With BAFT, the attention heatmaps *consistently focus on object-relevant regions (flower petals, center structures)*, even in shallow layers (Block 3) and deep layers (Block 11). This effect holds across all examples, not just for Bi-AdaptFormer or a single image.
>
> These results, combined with additional examples included in the supplementary material (Fig. 10-13 of Appendix A.11), confirm that **behavioral alignment improves representational focus throughout the network**.
>
> **2. The claim that lack of behavioral oversight is a limitation of PEFT is not well supported, as improvements are minor.**
>
> We would like to clarify two key points supported by our empirical results.
>
> **(i) Improvements are consistent across all PEFT methods and all datasets.** As shown in Table 1 of our main paper, BAFT improves Adaptformer, LoRA, Bi-LoRA and Bi-Adaptformer consistently. Across all 19 VTAB-1K datasets, BAFT provides consistent performance gains and never degrades performance. *This dataset-wide consistency is strong evidence of the role of behavioral drift in PEFT.*
>
> **(ii) VTAB-1K is widely regarded as a noisy, low-data benchmark where even +0.3–0.5% improvements are considered significant in the literature.** BAFT often achieves +1.0–2.0%, which is unusually large for a regularizer that introduces *zero new parameters*, and *is compatible with any PEFT method*.
>
> Therefore the observed improvements are not minor but rather meaningful and practically useful.
>
> In addition, the limitation we identify is behavioral misalignment, not merely accuracy.
>
> BAFT produces substantial improvements in internal behavior: (i) sharper and less fragmented attention maps (Figs. 1, 4),  (ii) better feature–prediction consistency and lower intra-class variance (Fig. 7-9 of Appendix), (iii) improved relational structure (Appendix A.2), (iv) no negative transfers across 19 VTAB datasets.
>
> In low-data VTAB-1k settings, where PEFT baselines already operate near capacity, gains are necessarily modest. Nevertheless, **BAFT achieves consistent improvements across 19 tasks (and 5 FGVC datasets) and four PEFT families without adding parameters, indicating practical value beyond numerical gain alone.**

---

> > ### Author Response · Authors · 2025-11-21
> >
> > **3. $L_\textbf{BAFT}$ does not match the evaluation metric; why is it not disturbing, and why is it better than simply optimizing cross-entropy?''}**
> >
> > BAFT is not intended to approximate the evaluation metric. Instead, it serves as a **geometric regularizer** that improves optimization under PEFT constraints.
> >
> > - Cross-entropy operates solely on outputs; PEFT modules, however, update only a small subset of parameters, leading to feature-prediction misalignment, noisy gradients, and unstable adaptation.
> >
> > - BAFT stabilizes this process by enforcing relational consistency ($S_z \approx S_p$), thereby producing smoother gradients, more coherent features, and more reliable inductive structure.
> >
> > - Similar to weight decay, label smoothing, self-distillation, or contrastive regularizers, $L_\textbf{BAFT}$ improves generalization without explicitly matching the evaluation metric.
> >
> > - Our gradient analysis (Sec. 3.2) shows that BAFT guides features along meaningful relational directions that cross-entropy alone cannot achieve in PEFT settings.
> >
> > **When $\lambda = 0$ (Eq. (3) in the main paper), the loss reduces to the standard PEFT objective, indicating that BAFT provides a more general formulation that augments the original loss with an additional regularization term.**
> >
> > Across all PEFT baselines, CE+BAFT (Eq. (3)) yields consistent accuracy improvements and better interpretability, confirming that BAFT is not a disturbing term but a beneficial regularizer. BAFT improves performance across a wide range of $\lambda$ (see Fig. 5), demonstrating that it does not disturb CE optimization unless lambda is extremely large (e.g., $\lambda=5$ or $10$).
> >
> > We appreciate the reviewer’s engagement with our work. BAFT provides a simple, parameter-free, teacher-free regularization that consistently improves both behavior and performance across multiple PEFT families and 26 tasks.
> >
> > We hope these clarifications address all concerns.

---

### Official Review · Reviewer_f3ms · 2025-11-01

**Soundness:** 2
**Presentation:** 3
**Contribution:** 2
**Rating:** 4
**Confidence:** 3

**Summary:**

This paper introduces Behavior-Aligned Fine-Tuning (BAFT), a parameter-free, teacher-free regularization method that improves parameter-efficient fine-tuning (PEFT) of large vision models. Instead of modifying model architecture or adding trainable parameters, BAFT aligns intermediate feature similarities with the relational structure in the model’s prediction space, using cosine similarity matrices as behavioral guidance. The authors provide both theoretical analysis, which shows BAFT as a spectral alignment process, and empirical evidence demonstrating consistent performance gains across VTAB-1k and few-shot fine-grained visual classification tasks.

**Strengths:**

1. The behavior-guided process is novel and intuitive.
2. The paper is well-written and organized, with a clear structure.

**Weaknesses:**

1. The improvements in emperical results seems small, which might raise questions aboy=ut the practical significance.
2. The method is tested only on ViT-based architectures and classification, maybe it is better to verify on some other applications, such as LLM.
3. How do you deal with the data with a large scale batch size?
4. Could you compare with other methods, such as manifold alignment or relational distillation?

**Questions:**

See weakness.

---

> ### Author Response · Authors · 2025-11-21
>
> We thank the reviewer for the thoughtful comments and for highlighting the novelty of behavior-guided alignment and the clarity of the paper.
>
> We address all concerns point-by-point below.
>
> **1. The improvements appear small; is practical significance limited?**
>
> Although the VTAB-1k average improvement is modest ($\sim$0.4\%), it is *highly consistent across all 19 tasks* and across *four distinct PEFT families* (LoRA, AdaptFormer, Bi-LoRA, Bi-AdaptFormer). **Achieving uniform gains in low-data PEFT is known to be challenging**.
>
> Importantly, BAFT improves every PEFT method (LoRA, AdaptFormer, Bi-Adaptformer and Bi-LoRA), showing that *the gain is systematic*, not random noise.
>
> Additionally, when compared under equal computational budget (baseline with extended training time), BAFT still provides higher accuracy (+0.2–0.3%).
>
> BAFT also provides substantial qualitative benefits: attention maps (Figs. 1, 4, and 10-13) show markedly improved localization and reduced fragmentation, indicating better internal geometric coherence even when accuracy gains are modest. BAFT never harms performance on any dataset and adds *no parameters* and $<3%$ computational overhead, making even small but consistent improvements practically meaningful.
>
> **2. Only ViT-based models and classification tasks are evaluated; what about other modalities such as LLMs?**
>
> - Our evaluation follows standard practice in PEFT literature: VTAB-1k and FGVC are the primary benchmarks used by LoRA, AdaptFormer, Bi-AdaptFormer, and Bi-LoRA, ensuring fair comparison.
>
> - We already demonstrate scalability to a substantially larger model, ViT-L/16 (Table 2), where BAFT consistently improves performance.
>
> - BAFT is architecture-agnostic, operating only on cosine similarities of features and predictions. Extending BAFT to LLMs is a promising and natural direction, but evaluating on the most widely used vision PEFT benchmarks provides the fairest and clearest assessment of its effectiveness.
>
> To further validate BAFT’s generality beyond vision models, we extend our experiments to the GLUE benchmark using the pretrained DeBERTaV3-base model. As shown in the table below, BAFT provides *consistent gains* across both P-Adapter and LoRA, improving the average performance in all cases. **Notably, BAFT enhances both sentence-level and token-sensitive tasks (e.g., CoLA, STS-B, QNLI), highlighting its robustness across diverse linguistic phenomena.**
>
> | Method        | #Param | CoLA  | STS-B | RTE   | MRPC  | SST-2 | QNLI  | Avg   |
> |-|-|-|-|-|-|-|-|-|
> | P-Adapter     | 1.18M  | 68.57 | 91.45 | 85.34 | 88.90 | 95.53 | 94.19 | 87.33 |
> | **+ BAFT**        | 1.18M  | **69.10** | **92.58** | **85.65** | **88.98** | **96.05** | **94.23** | **87.77** |
> | LoRA (r=8)    | 1.33M  | 67.96 | 91.63 | 84.65 | 90.23 | 95.36 | 94.01 | 87.31 |
> | **+ BAFT**        | 1.33M  | **68.30** | **91.95** | **84.73** | **92.56** | **95.98** | **95.02** | **88.09** |
>
> **3. How does BAFT handle very large batch sizes?**
>
> BAFT scales well in practice.
>
> - While similarity matrices grow quadratically with batch size, typical PEFT batch sizes (32-128) incur negligible overhead ($<0.05$ GB).
>
> - We support subsampled or blockwise computation of similarity matrices, exploiting the smoothness of relational kernels.
>
> - Our implementation uses in-place and fused operations for efficient memory use.
>
> Most PEFT pipelines already use moderate batch sizes due to dataset and memory constraints, making BAFT directly compatible with standard regimes. In addition, we conduct experiments on BiAdaptformer with BAFT using  different batch sizes.
>
> | Batch Size | CIFAR-100 Baseline | CIFAR-100 **+ BAFT** | Patch Camelyon Baseline | Patch Camelyon **+ BAFT** |
> |-|-|-|-|-|
> | 16         | 73.8                | **74.0**              | 86.4                      | **86.9**                    |
> | 32         | 74.1                | **74.3**              | 87.0                      | **88.2**                    |
> | 64         | 74.6                | **75.0**              | 87.3                      | **88.7**                    |
> | 128        | 75.1                | **75.4**              | 87.8                      | **89.0**                    |
>
> This demonstrates that BAFT remains effective across a wide range of batch sizes, confirming consistent performance improvements.

---

> > ### Author Response · Authors · 2025-11-21
> >
> > **4. Comparison with manifold alignment or relational distillation?**
> >
> > These methods are not directly comparable.
> >
> > - Manifold alignment typically aligns representations across *different domains or models*, whereas BAFT is a *self-supervised, single-model regularizer*.
> >
> > - Relational distillation requires a *teacher model* and aims to preserve its relational structure, while BAFT *generates relational targets dynamically from the model’s own predictions, without any teacher*.
> >
> > - BAFT enforces *spectral/kernel alignment (Theorem 1)*, a different objective from classical KD.
> >
> > - KD-style methods *require additional forward passes or teacher storage*, whereas BAFT introduces no parameters and negligible overhead.
> >
> > We will add a discussion elaborating these distinctions in the final revision.
> >
> > We appreciate the reviewer’s positive assessment of the clarity, structure, and intuitiveness of the method. BAFT provides a simple, teacher-free, parameter-free regularization that consistently improves four PEFT families across 26 tasks with strong theoretical motivation.
> >
> > We hope these clarifications fully address the reviewer’s concerns.

---

### Official Review · Reviewer_WV6S · 2025-11-02

**Soundness:** 3
**Presentation:** 3
**Contribution:** 2
**Rating:** 4
**Confidence:** 3

**Summary:**

The paper proposes behavior alignment fine-tuning (BAFT), targeted at parameter-efficient fine-tuning (PEFT) settings. The method uses an alignment loss that tries to match the representation similarity and final output similarity of sample pairs within a batch. The paper also conducts gradient and spectral analysis of the method. Experiments show improvement when used with existing PEFT methods on computer vision fine-tuning datasets using ViT. Ablation studies explore the impact of the loss coefficient lamba and where to apply the BAFT loss.

**Strengths:**

1. The idea of matching internal representation and output representation similarity is novel to my knowledge.
2. The experiments are relatively extensive within computer vision. Particularly it evaluates the combination of BAFT with various PEFT methods, and show improvement in all cases.
3. The paper analyzes the effect of the method from a mathematical perspective.
4. The paper conducts the most essential ablation studies of where and how much to apply the proposed loss.
5. BAFT adds minor overhead in training.

**Weaknesses:**

1. The proposed BAFT loss does not seem to have a particular applicability to PEFT settings. It could also be used for pre-training. This makes it unclear why the motivation of this method is in particular to PEFT.
2. All experiments conducted are in computer vision, and use the same ViT-B/16 as the pretrained backbone. This limits our understanding on how useful it is on other modalities/tasks, and whether the effectiveness could hold when the model is larger.
3. The idea of using intermediate representation to improve training is not entirely new. It dates back as early as FitNet (2014) in a knowledge distillation context. From this perspective, the conceptual contribution of this paper is limited.
4. The improvement overall is not too significant on VTAB (~0.4%).
(a) An analysis on whether this improvement is worth the additional effort of tuning lambda is in question. As figure 5 shows, the performance is quite sensitive to lambda and it may require many trials, and it may be specific to each dataset.
(b) A comparison on how this improvement compares to extending the training time in proportion to the overhead of BAFT is needed.

**Questions:**

Please see the weaknesses.

---

> ### Author Response · Authors · 2025-11-21
>
> We thank the reviewer for the constructive comments and for highlighting the novelty of aligning prediction- and feature-level similarities, the breadth of experiments, the mathematical analysis, and the low overhead of BAFT.
>
> Below we address all concerns point-by-point.
>
> **1. BAFT does not seem specific to PEFT; it could also be used for pre-training.**
>
> While BAFT is general in principle, it explicitly targets limitations unique to PEFT.
>
> - PEFT restricts parameter updates to low-dimensional subspaces, making representation–prediction mismatch far more severe than in full fine-tuning. Eq. (2) shows that *BAFT provides high-level geometric guidance that compensates for this restricted update space*.
>
> - The empirical attention maps (Figs. 1, 4, and 10-13) highlight that PEFT baselines suffer from diffuse or inconsistent attention, which BAFT systematically corrects.
>
> - **BAFT relies on reasonably meaningful predictions to define a relational structure; this assumption holds in PEFT but not during early pre-training when predictions are uninformative**.
>
> *BAFT is therefore especially well motivated and effective in PEFT settings.*
>
> Below, we provide the experimental comparisons of full fine-tuning and PEFT baselines, with and without BAFT, on VTAB-1k.
>
> | **Method**         | **Natural** | **Specialized** | **Structured** | **Average** | **Gains** |
> |-|-|-|-|-|-|
> | Full finetuning    | 76.1        | 83.4             | 42.1           | 67.2        | -         |
> | **+ BAFT**         | 76.3        | 83.6             | 42.5           | 67.5        | **+0.3**  |
> | AdaptFormer        | 82.4        | 86.0             | 62.4           | 76.9        | --        |
> | **+ BAFT**         | 82.5        | 86.2             | 63.1           | 77.3        | **+0.4**  |
> | LoRA               | 81.9        | 85.1             | 62.3           | 76.4        |           |
> | **+ BAFT**         | 82.1        | 85.3             | 62.9           | 76.8        | **+0.4**  |
> | Bi-AdaptFormer     | 82.5        | 85.6             | 62.9           | 77.0        | --        |
> | **+ BAFT**         | 82.7        | 86.5             | 63.2           | 77.5        | **+0.5**  |
> | Bi-LoRA            | 81.9        | 85.3             | 63.0           | 76.7        | --        |
> | **+ BAFT**         | 82.1        | 85.9             | 63.2           | 77.1        | **+0.4**  |
>
> BAFT provides marginal gains for full fine-tuning, but consistently larger gains under PEFT settings, supporting our PEFT-specific motivation.
>
> **2. Experiments are limited to vision and ViT-B/16; unclear generality or scalability.**
>
> Thank you. We clarify that (a) BAFT is already evaluated on a substantially larger backbone (ViT-L/16), where it *consistently improves* AdaptFormer, LoRA, Bi-AdaptFormer, and Bi-LoRA (Table 2). This directly demonstrates scalability to larger models. (b) BAFT is architecture-agnostic: it uses only cosine similarities of prediction vectors and intermediate features and makes no assumptions about visual structure. (c) VTAB-1k and FGVC represent the standard and most diverse benchmarks for PEFT in vision, ensuring comparability with prior work.
>
> Below we provide more evaluations.
>
> **ViT-Large, ImageNet-21k**
>
> | Method          | Natural | Specialized | Structured | Mean | #Params (M) |
> |-|-|-|-|-|-|
> | AdaptFormer     | 83.8    | 86.0        | 61.0       | 76.9  | 0.42         |
> | **+ BAFT**      | 83.9    | 86.4        | 61.6       | 77.3  | 0.42         |
> | LoRA            | 83.7    | 85.7        | 61.6       | 77.0  | 0.81         |
> | **+ BAFT**      | 83.8    | 86.2        | 62.0       | 77.3  | 0.81         |
> | Bi-AdaptFormer  | 84.0    | 86.2        | 61.4       | 77.2  | 1.60         |
> | **+ BAFT**      | 84.2    | 86.5        | 62.2       | 77.6  | 1.60         |
> | Bi-LoRA         | 83.2    | 85.0        | 61.7       | 76.6  | 3.17         |
> | **+ BAFT**      | 83.4    | 85.4        | 62.1       | 77.0  | 3.17         |
>
> **Swin-Base, ImageNet-21k**
>
> | Method          | Natural | Specialized | Structured | Average |
> |-|-|-|-|-|
> | AdaptFormer     | 82.5    | 87.0        | 60.7       | 76.7     |
> | **+ BAFT**      | 82.6    | 87.1        | 60.8       | 76.8     |
> | LoRA            | 82.3    | 86.7        | 61.9       | 77.0     |
> | **+ BAFT**      | 82.5    | 87.0        | 62.2       | 77.2     |
> | Bi-AdaptFormer  | 82.3    | 87.2        | 61.1       | 76.9     |
> | **+ BAFT**      | 82.4    | 87.4        | 61.4       | 77.1     |
> | Bi-LoRA         | 82.1    | 87.2        | 61.5       | 76.9     |
> | **+ BAFT**      | 82.3    | 87.3        | 61.9       | 77.2     |

---

> > ### Author Response · Authors · 2025-11-21
> >
> > **3. Representation-based supervision is not new (e.g., FitNets).**
> >
> > Our method differs fundamentally from FitNets and distillation.
> >
> > - BAFT is entirely **teacher-free**; the relational targets are generated **dynamically** from the model’s own predictions. Distillation and FitNets require a fixed teacher and cannot produce such closed-loop, self-consistent supervision.
> >
> > - BAFT aligns *batch-level relational structure* via Gram matrices, not one-to-one feature targets. Theoretical analysis (Theorem 1) shows that BAFT performs *spectral/kernel alignment* across the batch, which is **substantially different from classical feature matching**.
> >
> > - To our knowledge, BAFT is the first method to use prediction-space similarities to guide representation evolution in PEFT, addressing representation–prediction mismatch uniquely exacerbated by PEFT constraints.
> >
> > **4. Small improvements ($\sim$0.4\%); is tuning $\lambda$ worth it? Sensitivity?''**
> >
> > (a) BAFT yields improvements that are *consistent across all 19 VTAB datasets and across four distinct PEFT families*. Such consistency is rare on VTAB-1k, where different methods often help different subsets of tasks.
> >
> > (b) Contrary to the concern, BAFT is not sensitive: Fig. 5 shows stable performance across several orders of magnitude ($10^{-8}$ to $1.0$). Only extremely large values ($>1$) harm performance. A single default value works well for most tasks (e.g., $\lambda=1$).
> >
> > (c) BAFT adds only 0.02-0.05 GB memory and 10-15 ms/step (Fig. 6 in main paper), less than 3% overhead.
> >
> > (d) Simply extending training time cannot replicate BAFT: PEFT baselines already train for 300 epochs on VTAB and 100 on FGVC; additional epochs yield minimal gains and do not induce the *spectral alignment* BAFT provides.
> >
> > **5. Comparison to longer training proportional to overhead.**
> >
> > Baselines already train for the same number of epochs as in their original papers (LoRA, AdaptFormer, Bi-AdaptFormer). At this stage they are well into their convergence regime.
> >
> > More epochs increase compute substantially but *do not change the representation-prediction geometry*. BAFT provides qualitatively different guidance, not merely additional optimization.
> >
> > **6. Generality to broader tasks.**
> >
> > BAFT is modality-agnostic by design. We focus on the widely used VTAB-1k and FGVC PEFT benchmarks to ensure fair and reproducible comparison; extending BAFT to detection, segmentation, and non-vision modalities is promising future work.
> >
> > We have extended experiments on the GLUE benchmark using the pretrained DeBERTAV3-base model. Results are presented below.
> >
> > | Method        | #Param | CoLA  | STS-B | RTE   | MRPC  | SST-2 | QNLI  | Avg   |
> > |-|-|-|-|-|-|-|-|-|
> > | P-Adapter     | 1.18M  | 68.57 | 91.45 | 85.34 | 88.90 | 95.53 | 94.19 | 87.33 |
> > | **+ BAFT**        | 1.18M  | **69.10** | **92.58** | **85.65** | **88.98** | **96.05** | **94.23** | **87.77** |
> > | LoRA (r=8)    | 1.33M  | 67.96 | 91.63 | 84.65 | 90.23 | 95.36 | 94.01 | 87.31 |
> > | **+ BAFT**        | 1.33M  | **68.30** | **91.95** | **84.73** | **92.56** | **95.98** | **95.02** | **88.09** |
> >
> > We appreciate the reviewer’s positive assessment of the paper’s soundness, presentation, analysis, and breadth of experiments. BAFT is simple, efficient, teacher-free, and consistently improves four distinct PEFT families across 19 diverse tasks, supported by theoretical insights and ablations.
> >
> > We hope the clarifications provided address all concerns.

---

### Meta-Review · Area_Chair_MaSv · 2026-01-05

**Summary:**

The paper presents BAFT, a parameter-free method designed to align intermediate feature similarities with prediction similarities during PEFT.

Reviewers acknowledged the extensive experiments but expressed concerns regarding the practical relevance of the average performance gains (around 0.4%), the exclusive use of vision tasks for evaluation, and the method's differentiation from previous techniques such as distillation. In the rebuttal, authors addressed several of the raised issues. They demonstrated BAFT's effectiveness on NLP benchmarks (GLUE) and larger vision models (ViT-L, Swin), thereby broadening its applicability. The distinction from teacher-based distillation was also clarified.

That said, although improvements are consistent across 19 VTAB tasks, they remain modest. A further unresolved question is whether the consistent yet limited performance gain justifies the introduction of the additional hyperparameter \lambda, as indicated in Fig. 5(a) and (b).

**Reviewer Concerns:**

See above.

**Reviewer Scores:**

It's difficult to say. One reviewer might raise the score, but most will likely stick to their original stance.

---

### Decision · Program_Chairs · 2026-01-26

Reject